# The birth of a bacterial tRNA gene by large-scale, tandem duplication events

**Gökçe B Ayan[1], Hye Jin Park[1,2], Jenna Gallie[1]\***

[1]Department of Evolutionary Theory, Max Planck Institute for Evolutionary Biology, Plön, Germany; [2]Asia Pacific Center for Theoretical Physics, Pohang, Republic of Korea

**Abstract** Organisms differ in the types and numbers of tRNA genes that they carry. While the evolutionary mechanisms behind tRNA gene set evolution have been investigated theoretically and computationally, direct observations of tRNA gene set evolution remain rare. Here, we report the evolution of a tRNA gene set in laboratory populations of the bacterium *Pseudomonas fluorescens* SBW25. The growth defect caused by deleting the single-copy tRNA gene, *serCGA*, is rapidly compensated by large-scale (45–290 kb) duplications in the chromosome. Each duplication encompasses a second, compensatory tRNA gene (*serTGA*) and is associated with a rise in tRNA-Ser(UGA) in the mature tRNA pool. We postulate that tRNA-Ser(CGA) elimination increases the translational demand for tRNA-Ser(UGA), a pressure relieved by increasing *serTGA* copy number. This work demonstrates that tRNA gene sets can evolve through duplication of existing tRNA genes, a phenomenon that may contribute to the presence of multiple, identical tRNA gene copies within genomes.

**\*For correspondence:** gallie@evolbio.mpg.de

**Competing interests:** The authors declare that no competing interests exist.

## Introduction

Even though tRNAs perform the same canonical function in all organisms – decoding 61 sense codons into 20 amino acids – tRNA gene sets vary considerably across the tree of life (*Fujishima and Kanai, 2014*; *Marck and Grosjean, 2002*). Two aspects in which they vary are the types of tRNAs encoded and the number of gene copies encoding each type. Bacterial tRNA complements typically contain 28–46 types of tRNA, encoded by 28–120 genes (*Chan and Lowe, 2016*). Elucidating the factors contributing to, and the molecular mechanisms behind, the evolution of these variations has been of long-standing interest in biology. There is general agreement that bacterial tRNA gene sets are, in conjunction with the rest of the translational machinery, shaped by selection for rapid and accurate protein synthesis ('translational efficiency'; reviewed in *Gingold and Pilpel, 2011*). Efficient translation is an important determinant of bacterial growth rate, with more efficient protein production enabling faster growth and division (*Kurland, 1996*; *Kurland and Ehrenberg, 1987*). tRNAs mainly contribute to translational efficiency during elongation, the stage of translation where codons are sequentially matched to aminoacylated (charged) tRNAs (reviewed in *Gingold and Pilpel, 2011*; *Rodnina, 2018*). Codon-tRNA matching occurs by a trial and error process; tRNAs – in the form of ternary complexes (*Bensch et al., 1991*) – are stochastically sampled from the available pool. The speed with which a matching tRNA is selected depends on the absolute and relative concentration of tRNAs that match the codon; codons matched by more abundant tRNAs are expected, on average, to be translated more quickly than those matched by rarer tRNAs (reviewed in *Plotkin and Kudla, 2011*). Given that the formation of codon-tRNA matches by stochastic sampling is the rate-limiting step of elongation (*Varenne et al., 1984*), any factors affecting the matching process are likely to influence the evolution of bacterial tRNA gene sets. Examples include variations in codon-tRNA matching patterns and codon bias. Both of these are discussed in more detail below.

Codon-tRNA matching patterns are complex; some tRNAs match, and hence translate, more than one synonymous codon (*Crick, 1966*; *Ikemura, 1981*). This expanded translational capacity is the result of relaxed base pairing between the first base of the tRNA anticodon (tRNA position 34) and third base of the codon (codon position 3). In this binding position, a number of non-standard, 'wobble' pairings are permitted (reviewed in *Agris et al., 2018*). Most wobble pairings involve post-transcriptionally, enzymatically modified bases in the anticodon stem-and-loop region of tRNAs (*Boccaletto et al., 2018*; *Machnicka et al., 2016*). These post-transcriptional modifications affect the binding capacity and/or accuracy of a large fraction of bacterial tRNAs (*Björk and Hagervall, 2014*; *Manickam et al., 2016*). Hence, the set of post-transcriptional modification pathways active within a bacterial species affects the codon–tRNA matching pattern and, in turn, is expected to influence tRNA gene set composition. Indeed, various post-transcriptional modification pathways have been shown to co-vary with tRNA repertoires (*Diwan and Agashe, 2018*; *Grosjean et al., 2010*; *Novoa et al., 2012*).

Codon bias refers to the preferential use of some synonymous codons over others. A number of hypotheses exist regarding the evolutionary origins and consequences of preferred codons (reviewed in *Novoa and Ribas de Pouplana, 2012*; *Plotkin and Kudla, 2011*), one of which is the optimization of codon–tRNA matching during elongation (*Berg and Kurland, 1997*; *Bulmer, 1991*; *Bulmer, 1987*; *Higgs and Ran, 2008*; *Rocha, 2004*). There are several lines of support for this hypothesis. Firstly, different codons have been demonstrated to be translated at different rates in yeast, with more frequent codons generally being translated more quickly (*Gardin et al., 2014*). Secondly, codon and tRNA abundances have been observed to co-vary across many bacterial genomes (reviewed in *Ikemura, 1985*), particularly under conditions where rapid translation is required (e.g., during rapid growth [*Dong et al., 1996*; *Emilsson and Kurland, 1990*], in highly expressed genes [*Ikemura, 1981*], or among bacteria with faster growth rates [*Sharp et al., 2010*]). Thirdly, a number of studies have demonstrated an increase in protein expression when codon–tRNA co-variation is strengthened, either by optimizing synonymous codon use (*Sørensen et al., 1989*; *Zhou et al., 2004*) or by the addition of exogenous tRNAs (*Gu et al., 2004*; *Misra and Reeves, 1985*).

As outlined above, two factors expected to affect tRNA gene set evolution are the presence of tRNA post-transcriptional modification pathways and codon bias; post-transcriptional modifications heavily influence codon–tRNA matching patterns (affecting the tRNA types that are encoded), while codon bias dictates the translational demand for individual tRNA types (influencing tRNA abundances and hence tRNA gene copy number). Overall, theoretical and computational studies are consistent with the optimization of translational efficiency by the streamlining of tRNA gene sets; bacterial growth rate correlates with fewer tRNA types encoded by more gene copies (*Ran and Higgs, 2010*; *Rocha, 2004*).

In addition to factors influencing the evolution of tRNA gene sets, the mechanisms behind their evolution are an area of interest. Hypothetically, tRNA gene sets can evolve by several different mechanisms. Surplus tRNA genes may be lost (by deletion), while tRNA genes may be acquired from external sources (by horizontal gene transfer), or from within the genome (by duplication events). Additionally, the identity of existing tRNA genes may be altered by the acquisition of anticodon mutations (anticodon switching). Thus far, most evidence for the above routes of tRNA gene set evolution is indirect: phylogenetic analyses provide evidence of the flexibility of bacterial tRNA gene sets by loss, gain (both by horizontal gene transfer and duplication), and anticodon switch events (*Diwan and Agashe, 2018*; *McDonald et al., 2015*; *Tremblay-Savard et al., 2015*; *Wald and Margalit, 2014*; *Withers et al., 2006*). An anticodon switch event has also been directly observed in laboratory yeast; *Saccharomyces cerevisiae* populations in which the gene encoding tRNA-Arg(CCU) had been removed were repeatedly rescued by a C→T mutation in one of eleven gene copies encoding tRNA-Arg(UCU) (*Yona et al., 2013*). While the aforementioned study demonstrates the power of experimental evolution to provide insight into the evolution of tRNA gene sets, there remains a shortage of empirical studies directly investigating the evolution of bacterial tRNA gene sets and translation.

To address this, we (i) engineer a suboptimal bacterial tRNA gene set, (ii) compensate the defect using experimental evolution, and (iii) determine the genetic and molecular bases of compensation. More specifically, we delete the single-copy tRNA gene, *serCGA*, from the bacterium *Pseudomonas fluorescens* SBW25. We compensate the resulting growth defect during a 13-day serial transfer evolution experiment and show that the genetic basis of compensation is large-scale (45–290 kb),

tandem duplications encompassing a second tRNA gene (*serTGA*). Using a bacterial adaptation of YAMAT-seq (a method of mature tRNA pool deep-sequencing originally developed for use in human cell lines [*Shigematsu et al., 2017*]), we demonstrate that each duplication event is accompanied by an increase in tRNA-Ser(UGA) in the mature tRNA pool. Finally, we develop a model that combines our experimental results with the predicted effects of codon–tRNA matching patterns and codon bias, to provide a molecular explanation of how the observed tRNA pool changes may affect translation and growth.

## Results

### The *P. fluorescens* SBW25 tRNA gene set

Isolated from the leaf of a sugar beet plant, *P. fluorescens* SBW25 is a non-pathogenic bacterium that is frequently used as a model system in evolutionary biology. The SBW25 genome (*Silby et al., 2009*) is predicted by GtRNAdb 2.0 (*Chan and Lowe, 2016*) to contain 67 tRNA genes (*Figure 1A*; *Supplementary file 1*). The RNA product of one of these genes (*cysGCA-2*) is predicted to form a secondary structure that deviates significantly from the cloverleaf structure typical of canonical tRNAs (*Chan and Lowe, 2019*). Hence, the SBW25 tRNA gene set consists of 66 canonical tRNA genes. These encode 39 different tRNA types, 14 of which are encoded by multiple (between two and five) gene copies. Of these 14 types, 12 are encoded by gene copies that are identical in sequence; only tRNA-Asn(GUU) and tRNA-fMet(CAU) are encoded by multiple gene copies with different sequences.

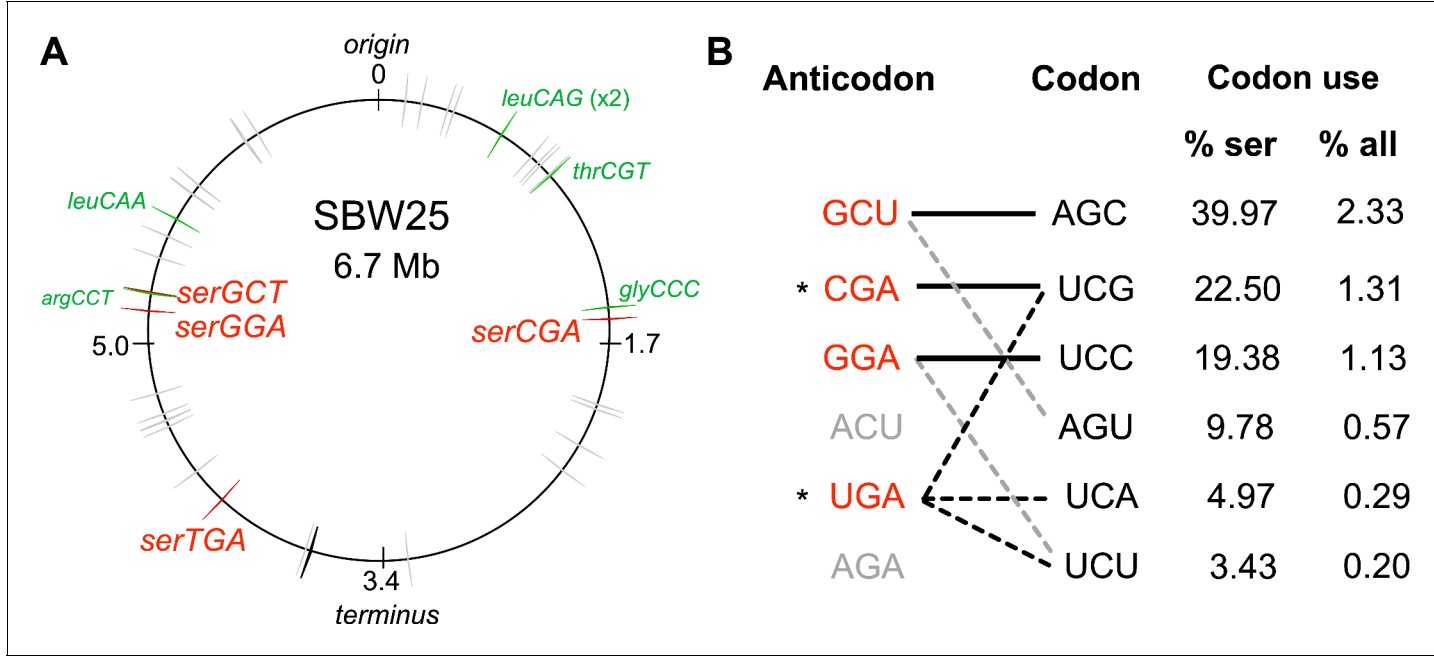

**Figure 1.** The tRNA gene set and serine translation system in *P. fluorescens* SBW25. (**A**) Genomic location of the 66 canonical (grey arrows) and one non-canonical (*cysGCA-2*; black arrow) tRNA genes. Four tRNA genes encode seryl-tRNAs (red arrows). One of these, *serCGA*, is predicted to encode a non-essential tRNA type. Six other tRNA genes encoding the remaining five non-essential tRNA types (green arrows). Replication origin and terminus are indicated. (**B**) The predicted translational relationship between seryl-tRNAs and serine codons. The six theoretically possible seryl-tRNA anticodons are listed on the left (red = present in SBW25, grey = absent, * = theoretically capable of translating codon UCG), and six cognate codons are listed in column 2. Connections signify a theoretical match (solid black lines = Watson Crick pairing; black dotted lines = wobble pairing through post-transcriptional modification; grey dotted line = G:U wobble pairing). Columns 3 and 4 list codon use as a percentage of serine and all codons, respectively (*Chan and Lowe, 2016*). Anticodons and codons are 5′→3′.

The online version of this article includes the following figure supplement(s) for figure 1:

**Figure supplement 1.** Predicted structure and function of tRNA types tRNA-Ser(CGA) and tRNA-Ser(UGA) in *P. fluorescens* SBW25.

The 39 different tRNA types identified in SBW25 must together be capable of translating all 61 sense codons. To investigate SBW25 codon–tRNA matching patterns, a combination of current wobble rules (reviewed in *Agris et al., 2018*) and tRNA post-transcriptional modification prediction tools (*Boccaletto et al., 2018*; *Machnicka et al., 2016*; *Panwar and Raghava, 2014*) was applied. Evidence for the translation of multiple synonymous codons was found for 28 of 39 tRNA types (*Supplementary file 1*). The proposed SBW25 codon–tRNA matching patterns indicate that the 39 tRNA types consist of a core set of 33 essential types (thatwhich together are theoretically capable of translating all 61 codons) and six non-essential types (whose cognate codons are also predicted to be translated by an essential tRNA type; *Figure 1A*; *Supplementary file 1*). The six non-essential types are candidates for suboptimal tRNA gene set construction; their apparent functional redundancy suggests that they can be eliminated, while their retention indicates that elimination may be detrimental.

With the aim of constructing a suboptimal tRNA gene set, we focussed our attention on *serCGA*, a single-copy gene encoding the non-essential tRNA type, tRNA-Ser(CGA) (*Figure 1A*). According to the predicted codon–tRNA matching pattern, the cognate codon of tRNA-Ser(CGA), UCG, can also be translated by the essential tRNA type, tRNA-Ser(UGA) (*Figure 1B*; *Supplementary file 1*). In several Gram-negative bacteria, tRNA-Ser(UGA) is post-transcriptionally modified at $U_{34}$ to 5-methoxycarbonylmethoxyuridine (mcmo$^5$U$_{34}$; *Boccaletto et al., 2018*), resulting in the expansion of translational capacity from codon UCA to include UCG and UCU (*Takai et al., 1999a*; *Takai, 1996*; see *Figure 1—figure supplement 1C*). The mcmo$^5$U$_{34}$ modification is performed by the CmoA/B/M enzymatic pathway (*Björk and Hagervall, 2014*; *Sakai et al., 2016*). Homologues of these enzymes are present in SBW25; protein BLAST searches of *Escherichia coli* MG1655 CmoA, CmoB, and CmoM against the SBW25 proteome give significant hits to Pflu1067, Pflu1066, and Pflu0633, respectively (BLASTp e-values <1e-50; *Altschul et al., 1990*).

CmoA/B/M-mediated expansion of tRNA-Ser(UGA) translational capacity to include codon UCG is expected to rescue a *serCGA* deletion mutant. Such a rescue event would require all UCG (and UCA) codons to be translated by tRNA-Ser(UGA). Given that UCG is a relatively high use codon in SBW25 – accounting for 22.5% of serine and 1.31% of all codons (*Figure 1B*) – *serCGA* deletion is expected to considerably increase translational demand for tRNA-Ser(UGA). Hence, while a *serCGA* deletion mutant may survive, significant translation and growth defects are anticipated.

## Deletion of *serCGA* limits rapid growth

To test the prediction that *serCGA* deletion results in a viable strain with a growth defect, the entire 90 bp, single-copy *serCGA* gene was removed from *P. fluorescens* SBW25 by two-step allelic exchange (see *Supplementary file 2* for construction details). This process changed the tRNA gene set from 66 tRNA genes of 39 types to 65 tRNA genes of 38 types. The engineering process was performed two independent times, yielding biological replicate strains Δ*serCGA*-1 and Δ*serCGA*-2. A third round of allelic exchange gave rise to an engineering control strain, SBW25-eWT. Successful deletion of *serCGA* demonstrates that it is not essential for survival; at least one other tRNA can translate codon UCG. *serCGA* deletion results in an immediately obvious growth defect on King's medium B (KB), a rich medium that supports rapid growth and hence rapid translation (*King et al., 1954*). Deletion of *serCGA* results in visibly smaller colonies on KB agar (*Figure 2A*). Compared with SBW25, Δ*serCGA*-1 shows a reduction maximum growth rate in liquid KB (two-sample *t*-test p=1.18×10$^{-6}$, *Figure 2B and C*) and elongated cells during growth in liquid KB (*Figure 2—figure supplement 1A*). Further, both independent *serCGA* deletion mutants lose when grown in direct, 1:1 competition with the neutrally marked SBW25-*lacZ* in liquid KB (one sample *t*-tests p<0.0001; *Figure 2D*). The observed growth defect is much less pronounced in minimal media, which supports slower growth and translation; SBW25 and Δ*serCGA*-1 colonies are similar sizes on M9 agar (*Figure 2E*), and no negative effects on growth or cell morphology were detected in liquid M9 (*Figure 2F and G*; *Figure 2—figure supplement 1B*). However, direct 1:1 competitions between Δ*serCGA* and SBW25-*lacZ* indicate a slight negative effect of *serCGA* deletion in liquid M9 (one sample *t*-tests p=0.06882 and 0.00908; *Figure 2H*).

The results in this section demonstrate that *serCGA* deletion leads to a growth defect that is more pronounced in KB than M9. The generation of a viable, but suboptimal, tRNA gene set upon *serCGA* deletion from SBW25 is consistent with the predictions of the previous section: that *serCGA* is not essential for translation, but it contributes to translational speed. It should, however, be noted

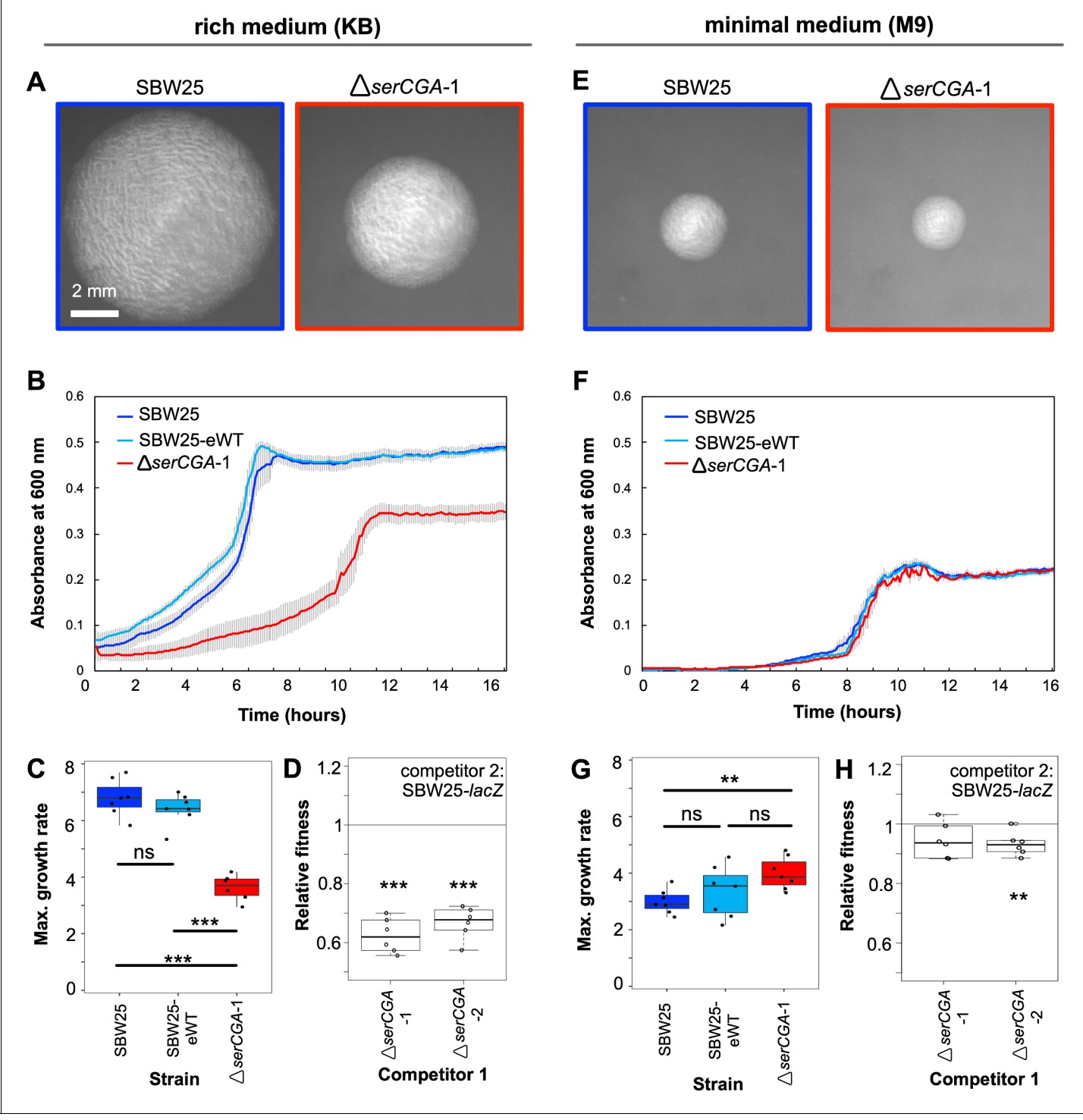

**Figure 2.** Deletion of *serCGA* leads to a growth defect in a rich medium. (**A**) 45 hr colonies on KB agar. (**B**) Growth (absorbance at 600 nm) in KB. Lines = mean of six (Δ*serCGA*-1) or seven (SBW25 and SBW25-eWT) replicates; error bars = one standard error. (**C**) Maximum growth rate (change in mOD min⁻¹) in KB, calculated with a sliding window of nine points between hours 2 and 12. (**D**) Relative fitness values from direct 1:1 competitions between competitor 1 (Δ*serCGA*-1 or Δ*serCGA*-2) and the neutrally marked wild-type strain, SBW25-*lacZ* (six replicates *per* competition), in KB. Relative fitness of 1 = no difference, <1 = SBW25-*lacZ* wins, >1 = competitor one wins. (**E**) 45 hr colonies on M9 agar, at same magnification and time as in panel A. (**F**) Growth in M9. Lines = mean of seven replicates, error bars = one standard error. (**G**) Maximum growth rate (change in mOD min⁻¹) in M9, calculated with a sliding window of nine points between hours 2 and 12. (**H**) Relative fitness values from direct 1:1 competitions between competitor 1

*Figure 2 continued on next page*

_Figure 2 continued_

(Δ_serCGA_-1 or Δ_serCGA_-2) and SBW25-_lacZ_ (six replicates _per_ competition), in M9. Parametric two-tailed two-sample _t_-tests (panel **C**, **G**) and parametric two-tailed one sample _t_-tests (panel **D**, **H**) \*\*\*p<0.001, \*\*p<0.01, \*p<0.05, ns = not significant (p>0.05).

The online version of this article includes the following source data and figure supplement(s) for figure 2:

**Source data 1.** This file contains the growth data used in _Figure 2_ panels B, C, F, and G.
**Source data 2.** This file contains the fitness data used to draw _Figure 2_ panels D and H, and _Figure 3_ panel E.
**Figure supplement 1.** The effect of _serCGA_ deletion on cell morphology during growth in liquid KB and M9.

that _serCGA_ deletion may increase intracellular serine levels, which could lead to toxic effects on growth. Indeed, significant increases in intracellular serine have been shown to affect the growth and division of _E. coli_ cells growing in rich, serine-containing media (**_Kriner and Subramaniam, 2020_**; **_Zhang et al., 2010_**; **_Zhang and Newman, 2008_**).

## The growth defect is repeatedly and rapidly compensated during experimental evolution

In order to investigate whether and how the growth defect exhibited by the _serCGA_ deletion mutant can be compensated for genetically, a serial transfer evolution experiment was performed. This experiment consisted of eight independent lineages: W1–W4 were control lines, each founded by a wild type strain (W1 and W2 by SBW25, W3 and W4 by SBW25-eWT), while M1–M4 were founded by the _serCGA_ deletion mutant (M1 and M2 by Δ_serCGA_-1, M3 and M4 by Δ_serCGA_-2). Each lineage was started from a single colony and maintained in liquid KB for 15 days, with daily transfer of 1% to fresh medium. Samples of each population were frozen daily.

After 13 days (~90 generations), all four mutant lineages (M1, M2, M3, and M4) showed visibly improved growth. Plating of day 13 populations on KB agar revealed that many colonies from lineages M1–M4 were larger than those of the founding _serCGA_ deletion mutant (**_Figure 3A_**). Notably, lineage M2 showed two phenotypically distinct types of large colonies: a phenotypically standard type and an opaque type. The opaque type closely resembles the previously reported switcher phenotype, in which on–off switching of colanic acid–based capsules generates opaque-translucent colony bistability (**_Beaumont et al., 2009_**; **_Gallie et al., 2019_**; **_Gallie et al., 2015_**; **_Remigi et al., 2019_**). No obvious change was observed in the size of colonies derived from day 13 of the four wild-type lineages (**_Figure 3A_**).

Five colonies were isolated from the day 13 mutant lineages for further analysis. These included one standard-looking, large colony from each mutant lineage (these isolates are hereafter referred to as M1-L, M2-L, M3-L, and M4-L) and a second large, opaque colony from lineage M2 (hereafter M2-Lop). Two representative colonies were isolated from day 13 of different wild-type lineages (hereafter W1-L and W3-L). Growth analyses in liquid KB showed improved growth profiles in all five isolates from the mutant lineages (**_Figure 3B–D_**). In line with the observed improvement, each of the five mutant lineage isolates outcompeted the _serCGA_ deletion mutant in direct, 1:1 competition (one sample _t_-tests p<0.01; **_Figure 3E_**). Indeed, no fitness difference was detected in 1:1 competitions between two mutant lineage isolates and neutrally marked SBW25-_lacZ_, providing evidence for high levels of compensation in these genotypes (M1-L and M4-L; one sample _t_-tests p>0.05; **_Figure 3E_**). The other three mutant lineage isolates were outcompeted by SBW25-_lacZ_, indicating partial compensation (M2-L, M2-Lop, and M3-L; one sample _t_-tests p<0.01; **_Figure 3E_**). No changes were observed in the growth or fitness of W1-L, the control isolate from day 13 of wild-type lineage 1 (**_Figure 3A–E_**).

The results in this section demonstrate that the growth defect caused by the deletion of _serCGA_ was repeatedly and rapidly compensated, to varying degrees, in isolates from each of the four mutant lineages on day 13 of the evolution experiment.

## Genetic basis of compensation is large duplications spanning _serTGA_

To determine the genetic basis of Δ_serCGA_ compensation, Illumina whole genome sequencing was performed on the seven day 13 isolates from the previous section: two control isolates from two

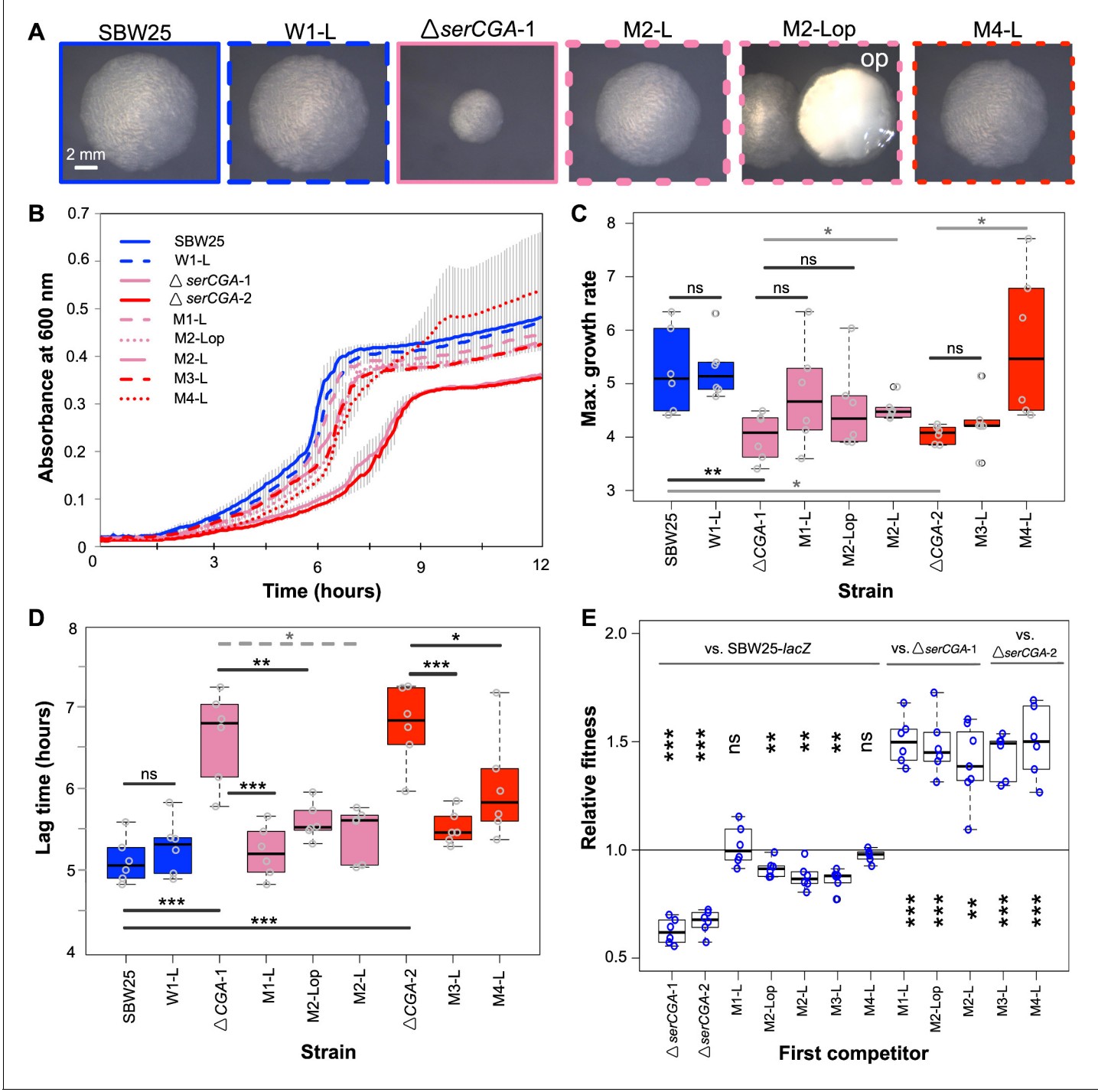

**Figure 3.** Loss of *serCGA* is repeatedly and rapidly compensated by experimental evolution. (**A**) Colony morphology of founder (solid outlines) and evolved (dotted outlines) isolates on KB agar (30 hr, 28°C). Lineage M2 yielded two large colony morphotypes: standard (left) and opaque ('op', right). Image border colours match line colours in panel B. (**B**) 12 hr growth curves in liquid KB for founder (day 0, solid lines) and evolved (day 13, dotted lines) isolates. Lines = mean of six replicates; error bars = 1 standard error. (**C, D**) Box plots showing the maximum growth rate (change in mOD min$^{-1}$) and lag time (hours) of founding and evolved strains from the evolution experiment, grown in liquid KB (n = 6; maximum growth rates and lag times calculated using a sliding window of nine points between 2 and 12 hr). Statistically significant differences were determined using parametric *t*-tests (solid black lines), non-parametric *t*-tests (solid grey lines), or Mann–Whitney–Wilcoxon rank sum tests (dotted grey line). (**E**) Box plots of the relative fitness of competitor 1 (*x*-axis) and competitor 2 (horizontal bars at top). Direct, 1:1 competitions were performed in liquid KB for 24 hr (28 °C, shaking). Six replicate competitions were performed for each set of strains. Relative fitness >1 means competitor one wins and <1 means competitor two wins.

*Figure 3 continued on next page*

*Figure 3 continued*

The first two competitions are also presented in *Figure 2G*. Statistically significant deviations of relative fitness from one were determined using parametric two-tailed one-sample *t*-tests. \*\*\*p<0.001, \*\*p<0.01, \*p<0.05, ns = not significant (p>0.05).

The online version of this article includes the following source data for figure 3:

**Source data 1.** This file contains the growth data used in *Figure 3* panels B, C, and D.

wild type lineages (W1-L and W3-L) and five isolates from four mutant lineages (M1-L, M2-L, M2-Lop, M3-L, and M4-L). In each of the five mutant lineage isolates a large, direct, tandem duplication was identified at around 4.16 Mb in the SBW25 chromosome (*Figure 4A–B*, *Figure 4—figure supplement 1*, *Supplementary file 3*). No evidence of any such duplications was found in either of the wild type control isolates. In addition to the large duplications, one synonymous point mutation was identified in one mutant lineage: in the carbohydrate metabolism gene *edd* of M2-L, codon 17 is changed from CGC to CGA (both encoding arginine; *Supplementary file 3*). This mutation was not identified in any other isolate, including the second isolate from the M2 lineage (M2-Lop), and is not considered likely to contribute to the compensatory phenotype of M2-L.

A combination of computational analyses, PCR, and Sanger sequencing was used to determine the precise region of duplication in four of five isolates. The duplications range in size from 45 kb (in M1-L) to 290 kb (in M4-L), and occur between 4.05 Mb and 4.34 Mb of the chromosome (*Table 1*). The precise location of the duplication in the fifth isolate (M3-L) could not be determined due to highly repetitive flanking DNA.

Next, we sought to identify the region(s) within the duplications responsible for the observed gain in fitness. Closer examination revealed a ~45 kb segment that is duplicated in all five strains (4,119,923–4,164,966). This segment is predicted to contain 45 genes, including 44 protein-coding genes (none of which is obviously linked to translation, or serine transport/metabolism), and one tRNA gene: *serTGA* (*Figure 4A*, *Supplementary file 4*). Close manual inspection revealed no evidence of any point mutations in any copy of the *serTGA* gene or promoter sequence, in any of the evolved isolates (*Supplementary file 5*), meaning that each duplication strain contains a second, wild-type copy of *serTGA*. In addition, isolate M4-L carries additional copies of four other tRNA genes (*argTCT*, *hisGTG*, *leuTAA*, and *hisGTG*; *Supplementary file 4*). Notably, this experiment has revealed three different tRNA gene sets that each provides a similar level of fitness in KB (see *Figure 3E*): SBW25 encodes 66 canonical tRNA genes of 39 tRNA types, four of five compensated isolates carry 66 tRNA genes of 38 types, and the fifth compensated isolate (M4-L) carries 70 tRNA genes of 38 types.

Given that the duplicated *serTGA* gene encodes tRNA-Ser(UGA), the tRNA type that can theoretically perform the function of tRNA-Ser(CGA) (see *Figure 1—figure supplement 1C*), it is a logical candidate for the underlying cause of compensation. However, there are also 44 protein-coding genes in the shared duplication segment, any of which could contribute to the compensatory effect. We therefore tested whether a plasmid-based increase in *serTGA* expression can compensate for *serCGA* loss. To this end, the *serCGA* and *serTGA* genes were individually amplified from SBW25, and each ligated into the expression vector pSXn (giving pSXn-CGA and pSXn-TGA; *Supplementary file 2*). This placed the expression of the tRNA gene under the control of an isopropyl-ß-D-1-thiogalactopyranoside (IPTG)-inducible *tac* promoter (*de Boer et al., 1983*; *Owen and Ackerley, 2011*).

Each plasmid construct was inserted into SBW25, Δ*serCGA*-1, and Δ*serCGA*-2, and the growth of the resulting strains was analysed in the absence of the inducer (to achieve lower-level, leaky expression of the tRNA gene). Expression of either *serCGA* or *serTGA* was shown to improve the growth of the *serCGA* deletion mutants in rich medium; addition of pSXn-CGA or pSXn-TGA increases the maximum growth rate of Δ*serCGA*, while addition of empty pSXn does not (*Figure 4C and D*; two-sample *t*-tests). Contrastingly, expression of neither tRNA improved growth of SBW25 (*Figure 4C*), with *serCGA* expression actually leading to a decrease in SBW25 maximum growth rate (*Figure 4D*; one-sided two-sample *t*-test p=0.000158). While it should be noted that tRNA-Ser(UGA) levels resulting from pSXn-based expression are likely to exceed those resulting from an additional

chromosomal copy of *serTGA*, the result that pSXn-based *serTGA* expression specifically improves the growth rate of Δ*serCGA* demonstrates that *serTGA* can provide a degree of compensation for *serCGA* loss. Other genes in the shared 45 kb fragment may nevertheless contribute to compensation, through unidentified mechanisms.

Thus far, our results provide strong evidence that the growth defect caused by *serCGA* deletion is repeatedly and rapidly compensated by one of several large-scale duplications encompassing *serTGA*.

## Large-scale duplications arise quickly and are heterogeneous

The duplication-carrying strains were isolated from all mutant lineages on day 13, demonstrating that the large-scale duplications occur repeatedly and rapidly. To more closely investigate the rapidity with which the duplications arose, a PCR was performed to identify the first time point at which the emergent M1-L duplication junction, M1junct1 (see *Supplementary file 2*), could be amplified from lineage M1 (see *Figure 4B*). The lineage M1 population samples frozen daily during the evolution experiment were revived and used as templates for the PCR. The M1-L duplication junction was first visible by PCR on day 3 and grew stronger as the experiment progressed (*Figure 4E*). Similar PCRs for the remaining duplication strains detected the presence of the relevant duplication junctions on day 2 (M2-Lop), day 4 (M3-L), and day 5 (M2-L and M4-L) (*Figure 4—figure supplement 2*).

A notable feature of the duplication fragments is that they are heterogeneous. That is, each of the five compensated isolates carries a unique duplication fragment with distinct endpoints (*Table 1*). Further, there is evidence of within-lineage heterogeneity; two distinct duplication fragments were identified in isolates from a single lineage (M2-L and M2-Lop; *Table 1*). To investigate within-lineage heterogeneity more closely, additional large colonies were isolated from day 13 of each mutant lineage and tested by PCR for the presence of the duplication junction(s) previously identified in the lineage. The results provide evidence for genotypic heterogeneity in compensated isolates from three of the four mutant lineages (M2, M3, and M4; *Figure 4—figure supplement 2*). Three differently sized PCR products were detected among the four isolates from lineage M2, demonstrating the presence of at least three distinct duplication fragments in the day 13 population. A mixture of compensated genotypes was also detected in lineages M3 and M4, with duplication junctions M3junct1 and M4junct1 amplifying in only three (of four) and two (of four) isolates, respectively. No PCR products were detected for the remaining isolates in these lineages, indicating that these three isolates either carry a duplication junction that was not tested for or compensate by a different mechanism.

The results in this section demonstrate that (i) strains carrying duplication fragments arise early within the mutant lineages of the evolution experiment (within 2–5 days or ~7–35 generations) and (ii) a degree of heterogeneity exists in duplication fragments, both between and within mutant lineages. These observations suggest that a mixture of duplication strains – and, by extension, tRNA gene sets – arise and compete within each mutant lineage.

## Duplication events increase the proportion of tRNA-Ser(UGA) in the mature tRNA pool

Next, we sought to quantify the effect of *serCGA* deletion and subsequent *serTGA* duplication on the mature tRNA pool of SBW25. To this end, YAMAT-seq – an established method of deep-sequencing mature tRNA pools in human cells (*Shigematsu et al., 2017*) and plants (*Warren et al., 2020*) – was adapted for use in *P. fluorescens* SBW25. The YAMAT-seq procedure quantifies charged and uncharged mature tRNAs; it does not measure pre-tRNAs, or tRNA fragments. Briefly, the YAMAT-seq procedure involves (i) isolation of total RNA from exponentially growing cells, (ii) removal of amino acids from the charged fraction of mature tRNAs, rendering all (or, most) mature tRNAs uncharged, (iii) ligation of Y-shaped, DNA/RNA hybrid adapters to the conserved, exposed ends of the mature, uncharged tRNAs, (iv) reverse transcription and amplification of adapter-tRNA complexes, (v) gel purification of the PCR products, (vi) high throughput sequencing, and (vii) computational and statistical analyses.

YAMAT-seq was performed on three replicates of nine strains: wild type (SBW25), the two independent *serCGA* deletion mutants (Δ*serCGA*-1, Δ*serCGA*-2), and six isolates from day 13 of the evolution experiment (W1-L, M1-L, M2-L, M2-Lop, M3-L, and M4-L). High throughput sequencing of the

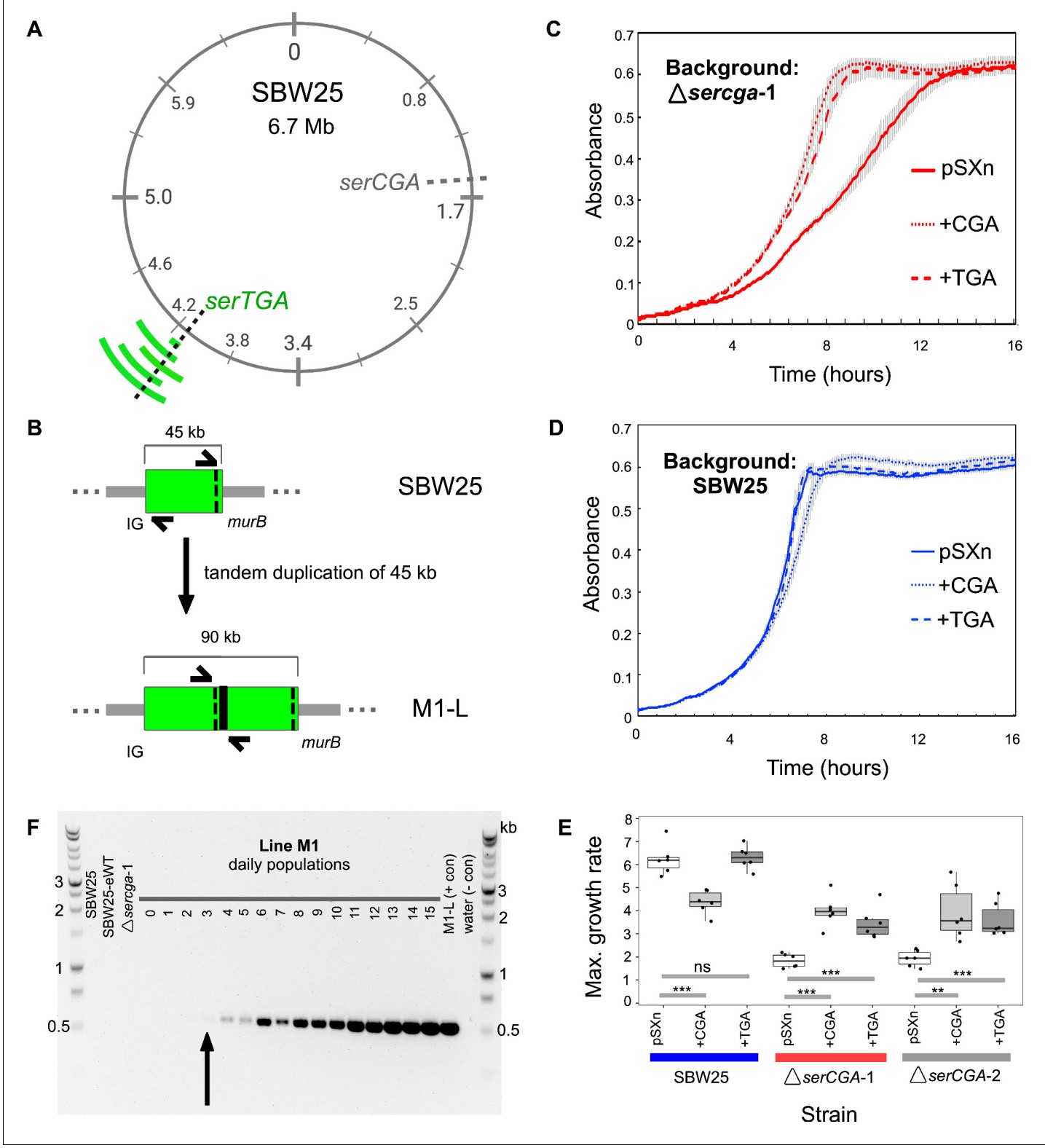

**Figure 4.** Direct, tandem duplications spanning *serTGA* compensate for *serCGA* loss. (**A**) Five isolates from the mutant Lines have unique, large, tandem duplications between 4.05 and 4.34 Mb of the SBW25 chromosome (green arcs; moving outwards: M1-L, M2-L, M2-Lop, M3-L, and M4-L). The duplications contain a shared 45 kb region with *serTGA* (dotted black line; see also **Figure 4—figure supplement 1**). (**B**) Cartoon depiction of the duplication event in M1-L, resulting in two copies of a 45 kb fragment (green) and an emergent junction (thick black line). The junction can be PCR-amplified using primers to either side (black arrows). IG = intergenic, black dotted line = *serTGA*. (**C**) 12 hr growth curves in LB+Gm (20 μg ml$^{-1}$) for

*Figure 4 continued on next page*

*Figure 4 continued*

ΔserCGA-1 (red) and SBW25 (blue) expressing *serCGA* or *serTGA* from the pSXn plasmid. Lines = mean of six replicates, error bars = 1 standard error. (D) Maximum growth speed (change in mOD min$^{-1}$; calculated with a sliding window of points between 0 and 23 hr) of SBW25, ΔserCGA-1, ΔserCGA-2 carrying empty pSXn, pSXn-*serCGA* (+CGA) and pSXn-*serTGA* (+TGA). Parametric two-tailed, two-sample *t*-tests ***p<0.001, **p<0.01, *p<0.05, ns = not significant (p>0.05). (E) The duplication junction in lineage M1 was first definitively amplified from lineage M1 population on day 3 (black arrow). Gel photograph colours were inverted using Preview to better detect faint PCR products. See *Figure 4—figure supplement 2* for the history of other junctions.

The online version of this article includes the following source data and figure supplement(s) for figure 4:

**Source data 1.** This file contains the growth data used in *Figure 4* panels C and D.

**Figure supplement 1.** Coverage plots from whole genome sequencing data provide evidence of large-scale, tandem duplication events in evolutionary lineages M1–M4.

**Figure supplement 2.** Large tandem duplications are detected between days 2 and 5 of the evolution experiment.

reverse-transcribed tRNA pools resulted in an average of 1,177,340 raw reads *per* sample. More than 99.99% of the raw reads fell within the range of lengths expected for tRNA-containing reads (80–150 bp), indicating good adapter-binding specificity. The >80 bp reads for each sample were aligned to a set of 42 reference tRNA sequences, consisting of all unique tRNA gene sequences predicted in the SBW25 chromosome (including *cysGCA*-2; *Supplementary file 6*; *Chan and Lowe, 2016*). At the conclusion of the alignment process, an average of 1,050,749 reads *per* sample were aligned to the reference sequences (~89% read retention; *Supplementary file 7*). Within each sample, the reference sequences with the highest and lowest (above zero) read counts consistently varied by a factor of ~10,000. For example, in sample 1, 109,963 reads aligned to Gly-GCC and 13 to Ile2-CAU (*Supplementary file 7*).

All samples showed reads aligned to 40 or 41 (of 42) reference sequences. The reference sequences without reads were Ser-CGA (in Δ*serCGA* and derivatives, as expected) and Cys-GCA-2 (in all 27 samples). Together with the prediction of a non-standard secondary structure for *cysGCA-2* (*Chan and Lowe, 2019*; *Chan and Lowe, 2016*), our inability to detect the Cys-GCA-2 sequence in any sample strongly indicates that Cys-GCA-2 does not contribute to translation. Overall, the YAMAT-seq results support the tRNA gene set predicted for SBW25: all 39 predicted types of mature tRNAs were detectable, including the 33 essential and the six non-essential tRNA types (see *Figure 1A*, *Supplementary file 1*).

While the YAMAT-seq results provide a useful overview of the relative abundances of tRNAs in a mature tRNA pool, within-strain comparisons of different tRNA types should be interpreted cautiously. As outlined by *Shigematsu et al., 2017*, variations in tRNA structural components and post-transcriptional modifications can adversely affect the relative efficiency of the reverse transcription reaction for some tRNA types, reducing their apparent proportions. In our results, three tRNA types consistently align a very low proportion of reads (<0.0001): Phe-GAA, Glu-UUC, and Ile2-CAU. A much higher proportion of reads was expected in particular for Phe-GAA and Glu-UUC, both of

**Table 1.** Duplication junctions in five isolates from the mutant lineages reveal duplication fragments of 45–290 kb.

Base positions refer to the SBW25 wild type genome sequence (*Silby et al., 2009*). For a list and details of the genes contained within each duplication segment, see *Supplementary file 4*.

| Strain name | Dup. size (bp) | Junction name | Junction side 1 | | Junction side 2 | |
|---|---|---|---|---|---|---|
| | | | Base | Region | Base | Region |
| M1-L | 45,043 | M1junct | 4,164,966 | *murB* | 4,119,923 | Intergenic repeat |
| M2-L | 191,833 | M2junct1 | 4,310,940–4,311,029 | Intergenic repeat | 4,119,235–4,119,352 | Intergenic repeat |
| M2-Lop | 182,877 | M2junct2 | 4,224,306 | *nuoL* | 4,042,455 | *pflu3649* |
| M3-L | ~192,000 | M3junct1 | ~4,310,800 | Intergenic repeat | ~4,119,100 | Intergenic repeat |
| M4-L | 290,335 | M4junct1 | 4,339,314 | Intergenic | 4,048,979 | *pflu3655* |

which are the sole tRNA types responsible for decoding all synonymous codons for their respective amino acids (accounting for 3.66% and 5.43% of genome wide codons, respectively; *Chan and Lowe, 2016*). Given their low read numbers, and non-central role in our experiment, these three tRNA types were removed from downstream statistical analyses.

The strength of the YAMAT-seq procedure lies in comparing changes in mature tRNA levels across genotypes. Any issues with efficiencies are expected to remain relatively constant across the strains in this experiment, allowing changes in relative tRNA type abundances to be detected. To this end, DESeq2 (*Love et al., 2014*) was used to compare normalized expression levels of 36 mature tRNA types – all SBW25 tRNA types except for Phe-GAA, Glu-UUC, and Ile2-CAU – in pairs of strains. Firstly, the effect of deleting *serCGA* was investigated by comparing tRNA sequences from each of the two independent *serCGA* deletion strains with those in SBW25 (*Figure 5A*). The absence of tRNA-Ser(CGA) from both deletion mutants demonstrates that (i) tRNA-Ser(CGA) is encoded solely by the deleted *serCGA* gene, and (ii) under the conditions tested, tRNA-Ser(CGA) is indeed a non-essential tRNA type in SBW25 (i.e., can be eliminated without causing death). The *serCGA* deletion mutants also showed consistently lower levels of tRNA-Thr(CGU) (DESeq2 adjusted p<0.001), a result that may reflect a close metabolic relationship between threonine and serine (*Sawers, 1998*). No significant differences were detected between the two deletion mutants (DESeq2 adjusted p>0.1).

Next, the effect of evolution on the *serCGA* deletion mutant was investigated by pairwise comparisons between each day 13 isolate and its corresponding ancestor (*Figure 5B*). Importantly, no differences in tRNA pools were detected in the wild type control lineage (W1-L versus SBW25; DESeq2 adjusted p>0.1). Contrastingly, a single consistent, statistically significant difference was observed across the five mutant lineage isolates: the level of tRNA-Ser(UGA) was 2.06- to 2.60-fold higher than in the ΔserCGA ancestor (DESeq2 adjusted p<0.0001). A number of other tRNA types co-vary with the rise in tRNA-Ser(UGA) (*Figure 5B*). While none of these differences is statistically significant in all strains, they are consistently in the same direction (i.e., increase or decrease). Thus, while the main effect of *serCGA* deletion and *serTGA* duplication is the loss of tRNA-Ser(CGA) and elevation of tRNA-Ser(UGA) respectively, other more subtle effects are likely to exist. Finally, pairwise comparisons between the duplication strains reveal some tRNA pool differences between the different duplication strains (*Figure 5—figure supplement 1*). Interestingly, no differences were detected between strain M4-L and any of the other duplication strains (DESeq2 adjusted p>0.3), indicating that the four additional tRNA genes duplicated in M4-L do not contribute to the mature tRNA pool. It is possible that the duplication junction in M4-L – which lies 112 bp upstream of the duplicated *argTCT-hisGTG-leuTAG-hisGTG* tRNA genes – truncates a promoter, leading to little or no expression of the duplicated copies. This result highlights that tRNA gene copy number does not always correlate with mature tRNA level.

The YAMAT-seq data shows the major effects of our engineering and evolution on the mature tRNA pool of SBW25 (*Figure 5C*). In the wild type strain, the proportion of tRNA-Ser(CGA) is around 2.5-fold higher than that of tRNA-Ser(UGA) (0.015 and 0.0059, respectively). The dominant effect of *serCGA* deletion is elimination of tRNA-Ser(CGA), with the proportions of the other tRNA types, including tRNA-Ser(UGA), remaining relatively stable. The subsequent large-scale duplications spanning *serTGA* generate an approximately twofold increase in the relative abundance of tRNA-Ser (UGA).

## A model for how elevation of tRNA-Ser(UGA) increases translational efficiency

Thus far, our results show that the growth defect caused by tRNA-Ser(CGA) elimination can be compensated by large-scale duplication events that serve to increase the proportion tRNA-Ser(UGA) in the mature tRNA pool. In this final section, we develop a model to provide a molecular explanation of how elevating tRNA-Ser(UGA) levels may compensate for tRNA-Ser(CGA) loss.

As described in the introduction, tRNA pool composition is an important determinant of translational speed. During elongation, the codon occupying the ribosomal A site is matched to a corresponding tRNA by stochastically sampling from the available pool of ternary complexes. Ternary complexes consist of a tRNA, an elongation factor (EF-Tu), and GTP (*Bensch et al., 1991*). Given that EF-Tu is a highly abundant protein that binds all correctly charged tRNA types with approximately equal affinity (*Louie et al., 1984*; *Schrader et al., 2011*), and uncharged tRNAs with several

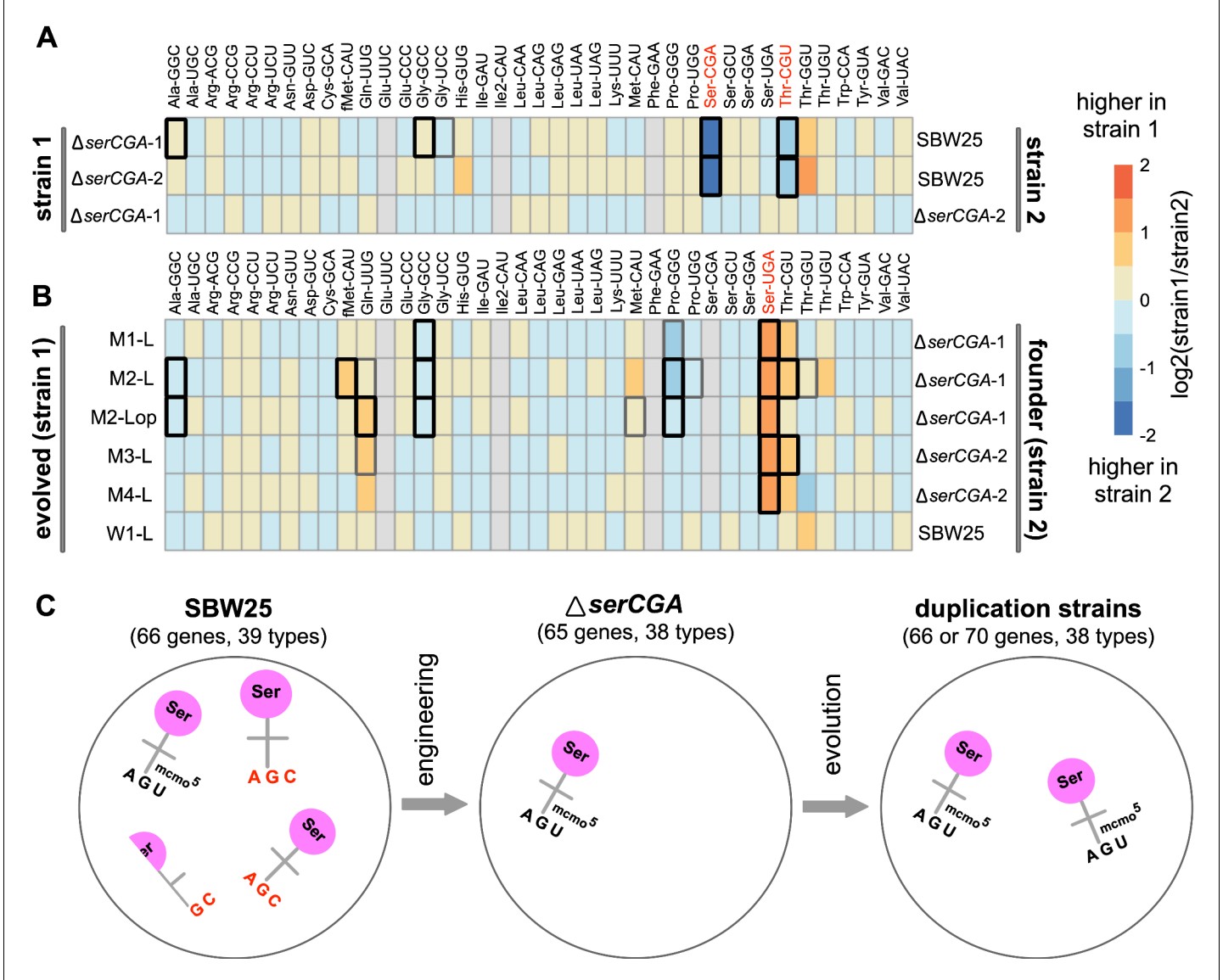

**Figure 5.** Heatmaps showing differences in mature tRNA levels between strains. The log2-fold.change(strain1/strain2) difference in expression for 36 mature tRNA types (with 38 different primary sequences) was determined for pairs of strains using DESeq2. (**A**) Mature tRNA expression levels in the *serCGA* deletion mutants compared with SBW25, demonstrating a consistently lower levels of tRNA-Ser(CGA) and tRNA-Thr(CGU) upon deletion of *serCGA*. tRNAs in red show a consistent difference in all comparisons except Δ*serCGA*-1 versus Δ*serCGA*-2 (row 3), the control comparison for which no significant differences in mature tRNA levels were detected. (**B**) tRNA-Ser(UGA) is higher in the mature tRNA pool in each of the five *serTGA* duplication isolates compared with the deletion mutant (with no significant differences detected in the wild type control Line, row 6). Some tRNA types were removed from the DESeq2 analysis (filled grey boxes): Glu-UUC, Ile2-CAU, Phe-GAA, and – in some comparisons – Ser-CGA, consistently gave low read numbers. Box borders represent statistical significance: thin grey = adjusted p>0.01, thick grey = 0.01 > adjusted p>0.001, black = adjusted p<0.001. tRNAs in red show a consistent difference in all comparisons except for the control (row 6). (**C**) Cartoon depicting the major effects of *serCGA* deletion (loss of tRNA-Ser(CGA)) and subsequent *serTGA* duplication (twofold increase of tRNA-Ser(UGA)) on the relative proportions of seryl-tRNAs in the mature tRNA pool.

The online version of this article includes the following source data and figure supplement(s) for figure 5:

**Source data 1.** This file contains the DESeq2 values from the tRNA expression analysis used in *Figure 5* panels A and B, and *Figure 5—figure supplement 1*.

**Figure supplement 1.** Comparison of expression levels of tRNA types in five strains isolated from mutant lineages on day 13.

**Table 2.** Numerical estimates of relative translation times using the mature tRNA pool measurements obtained during YAMAT-seq.

| Strain | YAMAT-seq proportion | | $\tau_{UCA}$ | $\tau_{UCG}$ | $\tau_{mRNA}$ | |
|---|---|---|---|---|---|---|
| | tRNA-Ser(UGA)[a] | tRNA-Ser(CGA)[b] | | | | $\tau_{mRNA}$ relative to tRNA gene set 1 |
| *tRNA gene set 1: wild type (one serCGA, one serTGA)* | | | | | | |
| SBW25 | 0.0059 | 0.015 | 169 | 47.8 | 385 | 0.970 |
| W1-L | 0.0056 | 0.014 | 179 | 51.0 | 408 | 1.03 |
| Mean | | | 174 | 49.4 | 397 | 1 |
| *tRNA gene set 2: serCGA deletion (0 serCGA, one serTGA)* | | | | | | |
| ΔserCGA-1 | 0.0050 | 0 | 200 | 200 | 1100 | 2.77 |
| ΔserCGA-2 | 0.0047 | 0 | 213 | 213 | 1170 | 2.95 |
| Mean | | | 207 | 207 | 1135 | 2.86 |
| *tRNA gene set 3: serCGA deletion and serTGA duplication (0 serCGA, two serTGA)* | | | | | | |
| M1-L | 0.013 | 0 | 76.9 | 76.9 | 423 | 1.07 |
| M2-Lop | 0.012 | 0 | 83.3 | 83.3 | 458 | 1.15 |
| M2-L | 0.012 | 0 | 83.3 | 83.3 | 458 | 1.15 |
| M3-L | 0.011 | 0 | 90.9 | 90.9 | 500 | 1.26 |
| M4-L | 0.010 | 0 | 100 | 100 | 550 | 1.39 |
| Mean | | | 86.9 | 86.9 | 478 | 1.20 |

fold lower affinity (**Nissen et al., 1996**; **Shulman et al., 1974**), the relative levels of each ternary complex are expected to reflect mature, charged tRNA proportions. Overall, the average time taken to match any given codon is dependent on the availability of the corresponding charged tRNA type (s); codons matched by a higher proportion of tRNAs will, on average, be translated more quickly than those matched by rarer tRNAs (**Varenne et al., 1984**). To illustrate, consider the serine codons and seryl-tRNAs from our experiments; codon UCA is translated by tRNA-Ser(UGA), while codon UCG can theoretically be translated by tRNA-Ser(CGA) *or* tRNA-Ser(UGA) (see **Figure 1B**). When both tRNA types are present, UCG is matched by a higher proportion of tRNAs than UCA and hence is expected to be translated more quickly, on average.

The above logic can be extended to provide relative numerical estimates of codon translation times, where codon-tRNA matching patterns and tRNA proportions are known. The average time required to translate any particular codon ($\tau_{codon}$) can be approximated by the inverse proportion of matching, charged tRNAs in the pool (**Equation 1**; where $p_{tRNA}$ is the proportion of matching tRNAs).

$$\tau_{codon} = \frac{1}{p_{tRNA}} \tag{1}$$

Accordingly, the average time taken to translate UCA can be estimated by the inverse proportion of tRNA-Ser(UGA) in the tRNA pool (**Equation 2a**), while UCG translation time can be estimated as the inverse proportion of tRNA-Ser(UGA) *plus* tRNA-Ser(CGA) (**Equation 2b**). For simplicity, these equations assume that tRNA-Ser(CGA) and tRNA-Ser(UGA) translate UCG with equal efficiency (see Discussion).

$$\tau_{UCA} = \frac{1}{p_{UGA}} \tag{2a}$$

$$\tau_{UCG} = \frac{1}{(p_{CGA} + p_{UGA})} \tag{2b}$$

If the proportions of mature tRNA-Ser(CGA) and tRNA-Ser(UGA) measured during YAMAT-seq are substituted into **Equations 2a and 2b**, relative measures of UCA and UCG translation times can be obtained in various genetic backgrounds (**Table 2**; see **Supplementary file 7**). According to these calculations, *serCGA* deletion increases the time taken to translate UCG by a factor of four

(mean UCG translation times of 49.4 and 207 in wild type and *serCGA* deletion tRNA gene sets, respectively). Subsequent *serTGA* duplication elevates the proportion of tRNA-Ser(UGA), partially restoring UCG translation time (mean UCG translation times of 207, 86.9, and 49.4 in the *serCGA* deletion strains, wild types, and *serTGA* duplication strains, respectively).

The cumulative impact of changing UCG translation speeds on translation and, ultimately, growth logically depends on how frequently the UCG codon is used. In SBW25, UCG is a relatively high use codon, occurring approximately 4.5 times more frequently than UCA (see *Figure 1B*; *Chan and Lowe, 2016*). To account for UCG codon bias, we estimated the average time required by each strain to translate an mRNA that reflects the relative UCG/UCA codon usage of SBW25. Using the same principle as in *Equation 1*, *Equation 2a and b*, the average time taken to translate the mRNA ($\tau_{mRNA}$) is calculated as the sum of the time taken to translate one UCA codon plus the time taken to translate 4.5 UCG codons (Equation 3).

$$\tau_{mRNA} = 1\left(\frac{1}{p_{UGA}}\right) + 4.5\left(\frac{1}{(p_{CGA} + p_{UGA})}\right) \tag{3}$$

The estimated average time required for the three tRNA gene complements to translate the mRNA can be calculated using the tRNA proportions measured during YAMAT-seq (*Table 2*). Using this method, on average a *serCGA* deletion mutant is expected to require approximately three times longer to translate the mRNA than the wild type (mean translation times of 1135 and 397, respectively). *serTGA* duplication is estimated to restore average translation time to near-wild type levels (mean translation times of 478 and 397, respectively).

Proportions of tRNA-Ser(UGA)[a] and tRNA-Ser(CGA)[b], as measured by YAMAT-seq for each strain (see *Supplementary file 7*), were substituted into *Equation 2a* to estimate UCA translation time ($\tau_{UCA}$), *Equation 2b* to estimate UCG translation time $\tau_{UCG}$, and *Equation 3* to estimate translation time of an artificial mRNA representing the relative codon use of UCA and UCG in SBW25 ($\tau_{mRNA}$). Each calculation is performed for the nine strains listed, which can be separated into three tRNA gene sets. Mean tRNA proportions and translation times are provided below the last strain in each tRNA gene set. tRNA proportions are given to two significant figures (s.f.) and calculated translation times to three s.f.

It should be noted that a limitation of the above model is that it assumes that all mature tRNAs are charged, while the YAMAT-seq proportions include both charged and uncharged mature tRNAs. The degree to which mature *E. coli* tRNAs are charged has been shown to vary with tRNA type and growth medium (*Avcilar-Kucukgoze et al., 2016*; *Dittmar et al., 2005*), with seryl-tRNAs demonstrating particularly low charging levels during exponential growth in rich medium (*Avcilar-Kucukgoze et al., 2016*). However, despite these differences, both previous studies report consistency of within-family tRNA charging levels. For example, during exponential growth in LB, all four *E. coli* seryl-tRNAs show a charging rate of ~10% (*Avcilar-Kucukgoze et al., 2016*). If similar, consistently low charging levels exist for SBW25 seryl-tRNAs during YAMAT-seq, our general conclusions are expected to hold given that the model uses only seryl-tRNA proportions to estimate elongation times.

Overall, the predictions of the model are consistent with our experimental results: the growth defect caused by *serCGA* deletion is compensated to near-wild type levels by the large-scale duplications encompassing *serTGA* (see *Figure 3*). Together, our results are consistent with the hypothesis that tRNA-Ser(CGA) elimination exerts increased translational demand on tRNA-Ser(UGA), and that this pressure is at least partially relieved by elevating tRNA-Ser(UGA) through increased *serTGA* copy number.

## Discussion

The evolutionary and molecular mechanisms by which different tRNA gene sets emerge have been of consistent, long-standing interest (periodically reviewed in *Gingold and Pilpel, 2011*; *Ikemura, 1985*; *Rak et al., 2018*). A multitude of theoretical studies have focused on various aspects of tRNA gene set evolution, highlighting roles for post-transcriptional modifications and codon bias (*Bulmer, 1987*; *Higgs and Ran, 2008*; *Ikemura, 1981*; *Novoa et al., 2012*; *Ran and Higgs, 2010*; *Rocha, 2004*; *Sharp et al., 2010*). Phylogenetic analyses have provided evidence of bacterial tRNA gene set evolution by four main routes: (i) tRNA gene loss through deletion events, (ii) tRNA gene

acquisition through horizontal gene transfer, (iii) tRNA gene acquisition by within-genome duplication events, and (iv) tRNA gene changes as a result of anticodon switching (*Diwan and Agashe, 2018*; *Marck and Grosjean, 2002*; *McDonald et al., 2015*; *Tremblay-Savard et al., 2015*; *Wald and Margalit, 2014*; *Withers et al., 2006*). In this work, we provide direct, empirical evidence for one of these routes: tRNA gene acquisition by within-genome duplication events. Loss of the single-copy tRNA gene *serCGA* was compensated by large-scale, tandem duplications that increase the copy number of tRNA gene *serTGA*, leading to elevation of tRNA-Ser(UGA) in the mature tRNA pool.

## Retention of *serCGA* in *P. fluorescens* SBW25 wild type

The observation that increasing the proportion of mature tRNA-Ser(UGA) can compensate for tRNA-Ser(CGA) loss can be explained on the molecular level: we hypothesize that elimination of tRNA-Ser (CGA) places translational strain on tRNA-Ser(UGA) at UCG codons, and that this pressure can be at least partially relieved by elevating tRNA-Ser(UGA) levels. While this hypothesis is consistent with our experimental results, it raises the question of why *P. fluorescens* SBW25 might retain a copy of *serCGA* in the natural, plant environment; given that selection for translational efficiency favours fewer tRNA types encoded by more tRNA gene copies (*Ran and Higgs, 2010*; *Rocha, 2004*), one might expect *serCGA* elimination in favour of multiple *serTGA* gene copies. However, *serCGA* is frequently retained in bacterial genomes; a study of 319 bacteria from different genera indicates a *serCGA* retention rate of ~70%, with retention correlating with higher UCG usage (*Wald and Margalit, 2014*). What is the advantage of retaining tRNA-Ser(CGA) over simply encoding higher levels of tRNA-Ser(UGA)?

One possible explanation for *serCGA* retention is that, while both tRNA-Ser(CGA) and tRNA-Ser (UGA) *can* translate UCG codons, tRNA-Ser(CGA) may do so with greater efficiency (i.e., more quickly and/or more accurately). Indeed, changing the anticodon of *E. coli* tRNA-Ser(UGA) from UGA to CGA has been shown to increase the efficiency of UCG translational in vitro (*Takai et al., 1999b*). Further, any difference in translational efficiencies may depend on environmental conditions; temperature, acidity, and ion concentration all alter the stability of RNA base pairings (*Nikolova and Al-Hashimi, 2010*; *Serra et al., 2002*). Overall more efficient translation of UCG by tRNA-Ser(CGA) in some environments could conceivably offset the cost of *serCGA* retention, particularly in bacteria with high UCG usage.

The possibility that restoration of *serCGA* may further increase the fitness of the *serTGA* duplication strains could be investigated by continuing the evolution experiment past day 13. This is because, in addition to increasing fitness, the duplication of *serTGA* has provided a possible route by which *serCGA* could be regained, and thus the original *P. fluorescens* SBW25 tRNA gene set restored. Specifically, one of the two copies of *serTGA* could acquire a T→C transition at tRNA position 34, changing the gene from *serTGA* to *serCGA* (i.e., an anticodon switch event). In order for such a mutation to spread through the population, it must encode a functional tRNA. This requires the new, hypothetical tRNA-Ser(CGA) to be recognized by seryl-tRNA ligase (SerRS), the enzyme that adds serine to all types of serine-carrying tRNAs in the cell (for a review of tRNA ligase function, see *Ibba and Soll, 2000*). Recognition of seryl-tRNAs by SerRS depends not on the tRNA sequence or the anticodon, but rather on the characteristic three-dimensional shape of seryl-tRNAs (*Lenhard et al., 1999*). Therefore, even though *serTGA* and the original *serCGA* encode tRNAs with very different sequences (see *Figure 1—figure supplement 1A*), it is plausible that the new, hypothetical *serCGA* could form a functional tRNA (see *Figure 1—figure supplement 1B*). Indeed, a UGA→CGA anticodon switch alters the translational capacity of *E. coli* tRNA-Ser(UGA) from codon UCA to UCG in vitro (*Takai et al., 1999a*; *Takai et al., 1999b*). Notably, none of the five genome sequenced, mutant lineage, day 13 isolates shows evidence of anticodon switch events in either copy of *serTGA* (see *Supplementary file 5*); whether anticodon switching occurs across a longer evolutionary time scale remains to be seen.

## Origin of large-scale duplications and new tRNA gene copies

Large-scale, tandem duplication events similar to those seen in this work are a well-documented adaptive solution to various phenotypic challenges in phage, bacteria, and yeast (reviewed in *Anderson and Roth, 1977*; *Elliott et al., 2013*; *Reams and Roth, 2015*). Extensive work has shown

that large-scale tandem duplications occur at extremely high rates in bacteria; for example, in an unselected overnight culture, around 10% of *Salmonella* cells reportedly carry a duplication of some sort, with 0.005–3% carrying a duplication of a particular locus (*Anderson and Roth, 1981*; *Anderson and Roth, 1977*; *Reams et al., 2010*). These rates are orders of magnitude higher than those typically reported for single nucleotide polymorphisms (*Westra et al., 2017*) and are consistent with the early detection of duplication fragments in our evolution experiment: strains carrying large-scale duplications were isolated from every mutant lineage by day 13 (~90 generations), and the duplication fragments they contained were first detected in the relevant population between days 2 and 5 (*Figure 4E*; *Figure 4—figure supplement 2*).

Large-scale duplications arise through unusual exchange of DNA between two separate parts of the bacterial chromosome, with the separating distance determining the size of the duplication (reviewed in *Anderson and Roth, 1977*; *Elliott et al., 2013*; *Reams and Roth, 2015*). A variety of mechanisms have been reported to underpin duplication formation, including (i) RecA-mediated, unequal recombination, (ii) RecA independent unequal recombination, and (iii) errant topoisomerase or gyrase activity (*Reams et al., 2014*; *Reams and Roth, 2015*; *Shyamala et al., 1990*). The first of these, RecA-mediated unequal recombination, occurs between direct repeats some distance apart (e.g., rRNA operons and *rhs* genes), with longer repeats generally leading to higher rates of recombination (*Anderson and Roth, 1981*). Two of the five mutant lineage isolates in this work (M2-L, M3-L) have endpoints in ~1.5 kb direct, imperfect repeats at 4.12 Mb and 4.31 Mb of the SBW25 chromosome, suggesting that the ~192 kb duplication fragment they contain arose by RecA-mediated, unequal recombination between these regions. The duplication fragments in the other three mutant lineage isolates (M1-L, M2-Lop, and M4-L) show no obvious signs of homology between their endpoints (*Table 1*), indicating an alternative mechanistic origin. Overall, the diversity in duplication fragment endpoints – location and degree of homology – in this study are indicative of a range of mechanistic origins.

While duplication formation does not necessarily require sequence homology, duplications that occur at the highest rates typically result from unequal recombination between long (>200 bp) repeats (*Anderson and Roth, 1981*). The *P. fluorescens* SBW25 genome contains many of these types of repeats dispersed around the chromosome, including five nearly identical rRNA operons and three highly similar *rhs* genes (*Silby et al., 2009*). In addition, there are hundreds of smaller repeats throughout the genome, including REPINs and tRNA genes (*Bertels and Rainey, 2011*; *Silby et al., 2009*). Errant recombination between any of these repeated sequences could plausibly generate large-scale duplications, meaning that almost any region of the SBW25 chromosome – and therefore many tRNA genes – could conceivably be duplicated *via* homologous recombination. It has previously been noted that the region surrounding the SBW25 replication terminus appears to be more susceptible to evolutionary change than the rest of the chromosome (*Silby et al., 2009*). This variable region extends approximately 1.4 Mb on either side of the terminus, engulfing 28 tRNA genes (including *serTGA*; see *Figure 1B* and *Supplementary file 4*). It seems probable that, while the copy number of many tRNA genes could feasibly be elevated by large-scale duplications, the tRNA genes surrounding the SBW25 replication terminus may be more prone to evolutionary change by duplication.

Given that dispersed repeats and large-scale duplications are a widespread feature of bacterial genomes (reviewed in *Brazda et al., 2020*; *Reams and Roth, 2015*), similar within-genome duplication may be capable of generating changes in tRNA gene copy number in many bacteria. Presumably, different duplication fragments – and therefore tRNA genes – arise at varying rates (depending on the presence and length of direct repeats in the surrounding area) and have varying degrees of evolutionary success (depending on fragment size and the dosage effects of genes in the fragment).

## Evolutionary fate of the large-scale duplications and new tRNA gene copies

Large-scale duplications in bacterial genomes are typically unstable (reviewed in *Reams et al., 2010*; *Reams and Roth, 2015*). That is, in addition to occurring at high rates, they are also lost – without a trace – at high rates. Duplications are lost in ~1% of cells *per* generation in Salmonella cultures, in the absence of selection (*Anderson and Roth, 1981*). Due to their combined ease of gain and loss, it has been suggested that duplications serve as fleeting evolutionary solutions to transient phenotypic challenges (*Sonti and Roth, 1989*); they occur at high frequency without adversely affecting

the long-term structure and integrity of the genome. The reported instability of large-scale duplications raises questions about the long-term fate of the duplications seen during this experiment. That is, if the serial transfer experiment was continued past day 13, what would happen to the duplication strains and the tRNA genes that they contain?

Extension of the evolution experiment could have several outcomes. One is that duplication fragments encompassing *serTGA* may continue to arise and be lost, with duplication strains eventually reaching a stable level within the population (*Reams et al., 2010*). In this scenario, while individual large-scale duplications would continually rise and fall, they would remain the dominant evolutionary solution in the population. A second possibility is that progressively smaller (and inherently more stable) duplication fragments may arise, perhaps eventually resulting in a duplication fragment comprised only of *serTGA* and its promoter. Such smaller duplication fragments may arise either from a *serCGA* deletion mutant (by rarer duplication events between more proximate DNA sequences) or from a duplication strain (by remodelling of the existing duplication fragment). Once a smaller, more stable fragment is dominant in the population, one copy of *serTGA* could conceivably change to *serCGA* through an anticodon switch event (see earlier discussion). A third possible outcome of continuing the evolution experiment is that, over time, a more stable mutation may arise, independently of the duplication fragments. Possible examples include (i) point mutation(s) in the promoter of *serTGA*, leading to increased *serTGA* expression without requiring additional gene copies, (ii) mutation(s) that extend the translational capacity of seryl-tRNAs to translate (or, to better translate) UCG codons, and (iii) synonymous point mutation(s) in highly expressed UCG codon(s), lowering the translational demand for tRNA-Ser(UGA). Should any of these more stable mutations arise, they would be expected to displace large-scale duplications – and in some cases, the additional *serTGA* copy – as the dominant evolutionary strategy in the population.

## Concluding remarks

The elimination of one tRNA type from *P. fluorescens* SBW25 was readily counteracted by large-scale duplication events that increased the gene copy number of a second, compensatory tRNA type. Together, our results provide a direct observation of the evolution of a bacterial tRNA gene set by gene duplication, and lend empirical support for the optimization of translation by codon–tRNA matching.

# Materials and methods

**Key resources table**

| Reagent type (species) or resource | Designation | Source or reference | Identifiers | Additional information |
|---|---|---|---|---|
| Gene (*P. fluorescens* SBW25) | *serCGA* | N/A | PFLUt39 | Encodes tRNA-Ser(CGA) |
| Gene (*P. fluorescens* SBW25) | *serTGA* | N/A | PFLUt51 | Encodes tRNA-Ser(UGA) |
| Strain, strain background (*P. fluorescens* SBW25) | *Pseudomonas fluorescens* SBW25 | *Rainey and Bailey, 1996*; *Silby et al., 2009* | | Wild type |
| Genetic reagent (*P. fluorescens* SBW25) | SBW25-*lacZ* | *Zhang and Rainey, 2007* | | Neutrally marked SBW25 for competition experiments |
| Genetic reagent (*P. fluorescens* SBW25) | Δ*serCGA*-1 | This work | | Bases 1624957–1625092, encompassing *serCGA*, removed. Biological replicate of Δ*serCGA*-2 |
| Genetic reagent (*P. fluorescens* SBW25) | Δ*serCGA*-2 | This work | | Bases 1624957–1625092, encompassing *serCGA*, removed. Biological replicate of Δ*serCGA*-1 |
| Genetic reagent (*P. fluorescens* SBW25) | SBW25-eWT | This work | | Wild type SBW25 that has been through the engineering process |

*Continued on next page*

*Continued*

| Reagent type (species) or resource | Designation | Source or reference | Identifiers | Additional information |
|---|---|---|---|---|
| Genetic reagent (*P. fluorescens* SBW25) | W1-L | This work | | Evolution isolate from day 13 of lineage W1 (founded by SBW25) |
| Genetic reagent (*P. fluorescens* SBW25) | W3-L | This work | | Evolution isolate from day 13 of lineage W3 (founded by SBW25-eWT) |
| Genetic reagent (*P. fluorescens* SBW25) | M1-L | This work | | Evolution isolate from day 13 of lineage M1 (founded by Δ*serCGA*-1) |
| Genetic reagent (*P. fluorescens* SBW25) | M2-L | This work | | Evolution isolate from day 13 of lineage M2 (founded by Δ*serCGA*-1) |
| Genetic reagent (*P. fluorescens* SBW25) | M2-Lop | This work | | Second evolution isolate from day 13 of lineage M2 (founded by Δ*serCGA*-1) |
| Genetic reagent (*P. fluorescens* SBW25) | M3-L | This work | | Evolution isolate from day 13 of lineage M3 (founded by Δ*serCGA*-2) |
| Genetic reagent (*P. fluorescens* SBW25) | M4-L | This work | | Evolution isolate from day 13 of lineage M4 (founded by Δ*serCGA*-2) |
| Recombinant DNA reagent | pSXn (plasmid) | *Owen and Ackerley, 2011*; Frederic Bertels | | pSX with one copy of a 38 bp direct repeat removed |
| Recombinant DNA reagent | pSXn-CGA (plasmid) | This work | | pSXn carrying *serCGA* |
| Recombinant DNA reagent | pSXn-TGA (plasmid) | This work | | pSXn carrying *serTGA* |
| Commercial assay or kit | DNeasy Blood and Tissue Kit | Qiagencat. no. 69506 | | |
| Commercial assay or kit | NextSeq 550 Output v2.5 kit | Illuminacat. no. 20024904 | | |
| Commercial assay or kit | TRIzol Max Bacterial RNA isolation kit | Life Technologiescat. no. 16096040 | | |
| Commercial assay or kit | DNA 7500 kit | Agilent Technologies | | |
| Software, algorithm | GtRNAdb 2.0 | *Chan and Lowe, 2016* | | https://www.gtrnadb.ucsc.edu |
| Software, algorithm | tRNAscan-SE 2.0 | *Chan and Lowe, 2019* | | https://www.lowelab.ucsc.edu/tRNAscan-SE/ |
| Software, algorithm | BLASTp | *Altschul et al., 1990* | | https://blast.ncbi.nlm.nih.gov/Blast.cgi?PAGE=Proteins |
| Software, algorithm | Gen5 | BioTek | | https://www.biotek.com/ |
| Software, algorithm | Geneious v11.1.4 | Geneious | | https://www.geneious.com/home/ |
| Software, algorithm | breseq v0.33.2 | *Deatherage et al., 2014a*; *Deatherage and Barrick, 2014b* | | https://barricklab.org/twiki/bin/view/Lab/ToolsBacterialGenomeResequencing |
| Software, algorithm | R v3.6.0 | *R Foundation for Statistical Computing, 2013* | | https://www.r-project.org/ |
| Other | SuperScript III reverse transcriptase | ThermoFisher Scientific cat.no.18080093 | | |
| Other | T4 RNA ligase 2 | New England BioLabs cat.no. M0239L | | Also known as T4 Rnl2 |
| Other | Phusion | ThermoFisher Scientific cat.no.M0531S | | |
| Other | 5% Mini-PROTEANTBE Gels | Bio-Rad Laboratories cat.no.4565015 | | |

## Strains, growth conditions, and oligonucleotides

Full lists of strains, plasmids, and oligonucleotides used are provided in *Supplementary file 2*. The *serCGA* deletion was constructed twice, independently; Δ*serCGA*-1 and Δ*serCGA*-2 are biological

replicates. Unless otherwise stated, *P. fluorescens* SBW25 cultures were grown in King's Medium B (KB; *King et al., 1954*) for ~16 hr at 28°C with shaking. *E. coli* strains were grown in Lysogeny broth (LB) for 16–18 hr at 37°C with shaking.

## Growth curves

Strains were streaked from glycerol stocks on KB, M9, or LB+Gm (20 µg ml$^{-1}$) plates. After 48 hr incubation, six or seven colonies *per* strain (numbers of replicates based on previous work: *Gallie et al., 2015*; *Lindsey et al., 2013*) were grown in 200 µl of liquid KB, M9, or LB+Gm (20 µg ml$^{-1}$) in a 96-well plate. Two microlitres of each culture were transferred to a fresh 198 µl of medium in a new 96-well plate, sealed with a plastic lid or a breathable rayon film (VWR), and grown at 28°C in a BioTek Epoch two plate reader. Absorbance at 600 nm (OD$_{600}$) of each well was measured at 5 min intervals, with 5 s of 3 mm orbital shaking before each read. Medium control wells were used to standardize other wells. The mean absorbance and standard error of the replicates at every time point were used to draw the growth curves in *Figures 2*, *3,* and *4*. Maximum growth rate and lag time were calculated using a sliding window of nine data points during the exponential growth window of the curve (Gen5 software from BioTek; see also source data files 1, 3, and 4).

## Fitness assays

For each competition, six replicates were performed in three separate blocks. All competitions within a block were performed in parallel. The number of replicates is based on previous work (*Beaumont et al., 2009*; *Gallie et al., 2019*). Single colonies of each competitor were grown independently in shaken KB. Competition tubes were inoculated with ~5 × 10$^6$ cells of each competitor and incubated at 28°C (shaking, 24 hr). Competitor frequencies were determined by plating on KB agar or LB+X-gal (60 µg ml$^{-1}$) agar at 0 and 72 hr. Competing genotypes were readily distinguished by their distinctive morphologies (on KB agar) or colour (neutrally marked SBW25-*lacZ* forms blue colonies on LB+X-gal; *Zhang and Rainey, 2007*). Relative fitness was expressed as the ratio of Malthusian parameters (*Lenski, 1991*) in *Figures 2* and *3*. Deviation of relative fitness from one was determined by two-tailed, parametric one-sample *t*-tests (see also source data file 2).

## Evolution experiment

SBW25 (wild type), SBW25-eWT (engineering control), and the two independent tRNA-Ser(CGA) deletion mutants (Δ*serCGA*-1 and Δ*serCGA*-2) were streaked from glycerol stocks onto KB agar and grown at 28°C for 48 hr. Two colonies from every strain were picked. Each of the eight chosen colonies became the founder of one evolutionary lineage. This resulted in four independent wild type lineages (W1–W4) and four mutant lineages (M1–M4).A medium control lineage was also included. The numbers of parallel lineages were chosen based on the available laboratory resources. Each colony was inoculated into 4 ml KB in a 13 ml plastic tube and incubated overnight at 28°C (shaking). Each grown culture (day 0) was vortexed for 1 min, and 100 µl was used to inoculate 10 ml KB in a 50 ml Falcon tube (28°C, shaking, 24 hr). Every 24 hr thereafter, 1% of each culture was transferred to a fresh 10 ml KB in a 50 ml Falcon tube, and a sample of the population frozen at −80°C. The experiment was continued until day 15. Populations were periodically dilution plated on KB agar to check for changes in colony size.

## Genome sequencing

Seven isolates were purified and stored from day 13 of the evolution experiment (W1-L, W3-L, M1-L, M2-L, M2-Lop, M3-L, M4-L). Genomic DNA was isolated from 0.5 ml overnight culture of each using a Qiagen DNeasy Blood and Tissue Kit. DNA quality was checked by agarose gel electrophoresis. Whole genome sequencing was performed by the sequencing facility at the Max Planck Institute for Evolutionary Biology (Ploen, Germany). Paired-end, 150 bp reads were generated with an Illumina NextSeq 550 Output v2.5 kit. Raw reads are available at NCBI sequence read archive (SRA accession number: PRJNA558233; *International Nucleotide Sequence Database Collaboration et al., 2011*). A minimum of 4.5 million raw reads *per* strain were aligned to the SBW25 genome sequence (NCBI genome reference sequence NC_012660.1; *Silby et al., 2009*) using breseq (*Deatherage and Barrick, 2014b*) and Geneious (v11.1.4). A minimum mean coverage of 94.7 reads *per* base pair was obtained. A full list of mutation predictions is provided in *Supplementary file 3*.

## Identification of duplication junctions

The duplication junctions in M1-L, M2-L, M2-Lop, M3-L, and M4-L were identified using a combination of analysis of whole genome sequencing data and laboratory-based techniques. The raw reads obtained from whole genome sequencing of each isolate were aligned to the SBW25 genome sequence (*Silby et al., 2009*) using breseq (*Barrick et al., 2014*; *Deatherage et al., 2014a*; *Deatherage and Barrick, 2014b*) and Geneious (v11.1.4). Coverage analyses were performed in Geneious, and coverage plots generated in R (v3.6.0) (*Figure 4—figure supplement 1*). Manual inspection of the Geneious alignment in coverage shift regions led to predicted junctions in all isolates except M3-L. Each predicted junction was checked by alignment to (i) raw reads and (ii) previously unused sequences (using Geneious). Junction sequences were confirmed by PCR and Sanger sequencing (for primer details, see *Supplementary file 2*).

## Historical junction PCRs

Glycerol stock scrapings of the frozen daily populations, or large colony isolates, from each mutant lineage were grown in liquid KB. Washed cells were used as PCR templates, alongside positive and negative controls (see *Supplementary file 2* for primer details). The PCR products were run on a 1% agarose gel against a 1 kb DNA ladder at 100 volts for 90 min. Gels were stained with SYBR Safe and photographed under UV illumination. In order to better detect faint PCR products in the earlier days of the evolution experiment, the colours in each photograph were inverted using Preview (v11.0) (*Figure 4E* and *Figure 4—figure supplement 2*).

## Expression of tRNA genes from the pSXn plasmid

Wild type copies of the *serCGA* and *serTGA* genes were individually ligated into the expression vector, pSXn, to give pSXn-CGA and pSXn-TGA. The pSXn vector contains an IPTG-inducible *tac* promoter (*de Boer et al., 1983*; *Owen and Ackerley, 2011*). Together with the empty vector, the two constructs were separately placed into the SBW25, ΔserCGA-1, and ΔserCGA-2 backgrounds. This was achieved by transformation of the vector constructs into chemically competent cells (*Gallie et al., 2015*). The growth profiles of the nine resulting genotypes were obtained in liquid LB +Gm (20 µg ml$^{-1}$), in six replicates of six (see Growth Curves methods, *Figure 4C and D*, and source data file 4). No IPTG was added at any stage, in order to achieve lower-level, leaky expression of the tRNA gene from the uninduced *tac* promoter.

## YAMAT-seq procedure

YAMAT-seq (*Shigematsu et al., 2017*) is adapted in this work for use in *P. fluorescens* SBW25. Three independent replicates (based on replicate numbers reported in *Shigematsu et al., 2017*) of nine strains (i.e., 27 samples) were grown to mid-exponential phase in 250 ml flasks containing 20 ml KB. Total RNA was isolated from 1.5 ml aliquots (TRIzol Max Bacterial RNA isolation kit). For each sample, 10 µg of total RNA was subjected to tRNA deacylation treatment – incubation in 20 mM Tris-HCl (pH 9.0) for 40 min at 37°C. Each deacylated RNA sample was desalted and concentrated by ethanol precipitation. Y-shaped, DNA/RNA hybrid adapters (Eurofins; *Shigematsu et al., 2017*) were ligated to the conserved, exposed 5'-NCCA-3' and 3'-inorganic phosphate-5' ends of uncharged tRNAs using T4 RNA ligase 2. Ligation products were reverse transcribed into cDNA using SuperScript III reverse transcriptase and amplified by 11 rounds of PCR with Phusion. One of the 27 sample-specific indices listed in *Supplementary file 7* was added to each of the 27 reactions. The quality and quantity of each PCR product were checked using an Agilent DNA 7500 kit on a Bioanalyzer, and samples combined in equimolar amounts into one tube. The mixture was run on a 5% native polyacrylamide gel, and the bands between 180 and 250 bp excised. DNA was extracted in deionized water overnight, and agarose removed by centrifugation through filter paper. The final product was sequenced at the Max Planck Institute for Evolutionary Biology (Ploen, Germany). Single-end, 150 bp reads were generated with an Illumina NextSeq 550 Output v2.5 kit. YAMAT-seq data is available at NCBI Gene Expression Omnibus (GEO accession number GSE144791) (*Edgar et al., 2002*).

## Analysis of YAMAT-seq data

Raw YAMAT-seq reads were sorted into 27 samples by extracting exact matches to each unique, 6 bp long Illumina index. Exact barcode matches were used in order to minimize the misallocation of reads as a result of barcode errors introduced during the sequencing process. The resulting 27 raw read files, each containing a minimum of 637,037 reads, were analysed using Geneious (version 11.1.4). Reads of the expected length (80–151 bp) were extracted (resulting in 99.99% read retention). The extracted reads were assembled to a set of 42 reference tRNA sequences from SBW25 (*Supplementary file 6*). During assembly, up to 10% mismatches, gaps of < 3 bp, and up to five ambiguities were allowed *per* read. Reads that aligned equally well to more than one reference sequence were discarded in order to minimize misallocation of reads to similar reference sequences. Finally, the unused reads for each sample were de novo aligned, and the resulting contigs checked to ensure that none contained substantial numbers of tRNA reads (particularly seryl-tRNAs). The above sorting and assembly parameters were initially based on those reported by *Shigematsu et al., 2017* and were subsequently refined for the SBW25 data. Following assembly, the within-sample proportion of reads aligned to each tRNA type was calculated, and mean mature tRNA proportions were calculated for each strain across the three replicates (*Supplementary file 7*). DESeq2 (*Love et al., 2014*) was used in R (version 3.6.0) to detect tRNA expression differences between pairs of strains (see source data file 5). DESeq2 corrects for multiple testing with the Benjamini–Hochberg procedure (*Anders and Huber, 2010*). Three tRNA types (Glu-UUC, Ile2-CAU, and Phe-GAA) were removed from the analyses due to very low read numbers (<0.01% of the total reads *per* strain).

## Statistical tests

Parametric two-tailed two-sample *t*-tests were performed to detect differences in maximum growth rate ($V_{max}$) and lag time in growth curves (*Figures 2C, G*, *3C, D*, and *4D*, source data files 1, 3, and 4) in cases where all assumptions were satisfied. Where equal variance or normality assumptions were violated, non-parametric Welch two-sample *t*-tests and Mann–Whitney–Wilcoxon rank sum tests were used, respectively (see *Figure 3C and D*, source data file 3). Parametric one-tailed one-sample *t*-tests were used to detect deviations of relative fitness values from one in competition assays (see *Figures 2D, H* and *3E*, source data file 2). DESeq2 adjusted (for multiple comparisons) p-values were used to detect differences in tRNA expression during YAMAT-seq (*Figure 5A and B*, *Figure 5—figure supplement 1*, *Figure 5—source data 1*). Analyses were performed in R (v3.6.0). Significance levels: ns = not significant ($p>0.05$), *$0.05 < p < 0.001$, **$0.01 < p < 0.001$, ***$p<0.001$.

## Acknowledgements

This work was supported by the Max Planck Society (all authors). The authors thank Gunda Dechow-Seligmann for technical assistance and Sven Künzel for assistance with troubleshooting the YAMAT-seq protocol. The authors also thank Frederic Bertels for discussions during development of the YAMAT-seq analysis, and the gift of the pSXn plasmid. The authors thank the anonymous reviewers, Frederic Bertels, Anuradha Mukherjee, Devika Bhave, and Bilal Haider, for their helpful comments on the manuscript.

## Additional information

### Funding

| Funder | Author |
|---|---|
| Max Planck Society | Gökçe B Ayan<br>Hye Jin Park<br>Jenna Gallie |

The funders had no role in study design, data collection and interpretation, or the decision to submit the work for publication.

## Author contributions
Gökçe B Ayan, Resources, Data curation, Software, Formal analysis, Validation, Investigation, Visualization, Methodology, Writing - review and editing; Hye Jin Park, Formal analysis, Visualization, Methodology, Writing - review and editing; Jenna Gallie, Conceptualization, Resources, Data curation, Software, Formal analysis, Supervision, Validation, Investigation, Visualization, Methodology, Writing - original draft, Project administration, Writing - review and editing

## Author ORCIDs
Gökçe B Ayan (ID) https://orcid.org/0000-0001-7999-0921
Hye Jin Park (ID) https://orcid.org/0000-0003-3552-6275
Jenna Gallie (ID) https://orcid.org/0000-0003-2918-0925

## Decision letter and Author response
Decision letter https://doi.org/10.7554/eLife.57947.sa1
Author response https://doi.org/10.7554/eLife.57947.sa2

## Additional files

### Supplementary files
• Supplementary file 1. Details of tRNA types predicted in the *P. fluorescens* SBW25 genome, and a putative codon-tRNA matching pattern. A list of 62 theoretically possible tRNA types (61 elongator tRNAs and one initiator tRNA) is provided. Of these, 23 are absent from the SBW25 tRNA gene set. Codon-tRNA matching patterns are based on the G-U wobble rule (*Crick, 1966*), and post-transcriptional modifications predicted using a combination of MODOMICS (*Boccaletto et al., 2018*), tRNA-mod (*Panwar and Raghava, 2014*), and tRNAmodpred (*Machnicka et al., 2016*). Following current wobble rules (described in *Agris et al., 2018*) and the predicted post-transcriptional modifications, the 39 tRNA types present in SBW25 are split into 33 theoretically essential types (pink highlighting) and six theoretically non-essential types (green highlighting). Gene copy numbers and genome-wide proportions of codon use (from GtRNAdb; *Chan and Lowe, 2016*) are listed for each tRNA type present. Further notes are provided where appropriate.

• Supplementary file 2. List of strains, plasmids, oligonucleotides, and duplication junctions used in this study.

• Supplementary file 3. Complete list of mutations predicted from the whole genome sequencing of isolates from day 13 of the evolution experiment. Whole genome sequence data (Illumina NextSeq, 150 bp, paired-end reads; SRA accession number: PRJNA558233) were obtained for seven strains from day 13 of the evolution experiment: W1-L, W3-L (each derived from two independent SBW25 control lineages), M1-L, M2-L, M2-Lop, M3-L, and M4-L (each derived from four independent *serCGA* deletion lineages). A minimum of 4.5 million raw reads was obtained for each sample, and these were aligned to the *P. fluorescens* SBW25 genome sequence (*Silby et al., 2009*) and subsequently analysed using a combination of breseq (*Barrick et al., 2014*) and Geneious. This file provides a summary and a full report of the mutations predicted in each isolate. In addition to the large tandem duplications detailed in *Table 1*, two unique point mutations were identified: W3-L carries a non-synonymous point mutation in *ytfH* (encoding a probable transcriptional regulator), and M2-L carries a synonymous point mutation in *edd* (encoding phosphogluconate dehydratase). There is evidence for some non-ubiquitous deletions in repetitive, intergenic, DNA elements in M1-L, M2-L, M2-Lop, and M4-L. These deletions affect areas outside of the large duplication regions. Some putative mutations were identified in all (or most) isolates, many of which have also been identified in unrelated experiments (i.e., are likely to be present in the starting strain; *Gallie et al., 2015*, *Gallie et al., 2019*). As such, these putative mutations are not expected to be relevant for the fitness effects described in this manuscript.

• Supplementary file 4. List of genes in each duplication fragment. The spreadsheet lists the SBW25 gene annotations from NCBI (6176 genes; left of the spreadsheet), followed by the details of which genes are duplicated in M1-L, M2-L, M2-Lop, M3-L, and M4-L. Note that for comparison purposes, the duplication details for each strain are provided on the same line numbers as the SBW25 list

(scroll down until the first duplicated genes is visible in each duplication isolate). The core set of 45 genes that is duplicated in each of the five isolates is highlighted in pink.

• Supplementary file 5. Whole genome sequencing base calls reveal no evidence of mutations in either copy of *serTGA* (or its promoter) in any of the five duplication-carrying strains. This file contains details of the raw read numbers of proportions of the dominant base called at SBW25 chromosome positions 4,163,616–4,163,861 in strains W1-L, W3-L (each carrying one *serTGA* copy), M1-L, M2-L, M2-Lop, M3-L, and M4-L (each carrying two *serTGA* copies). This 245 bp segment encompasses the *serTGA* gene and ~155 bp of the upstream region, which is expected to contain the *serTGA* promoter. Any point mutation in either *serTGA* copy in the duplication strains (M1-L, M2-L, M2-Lop, M3-L, and M4-L) is expected to be reflected by a drop in the proportion of the dominant base (to around 0.5). For example, if in M1-L one *serTGA* copy had gained a C→T point mutation in tRNA position 34 (i.e., an anticodon switch event; see Discussion), one would expect approximately half the 214 reads covering base 4,163,669 to contain a T, and the other half to carry a C. Therefore, the dominant base proportion would be expected to drop to ~0.5. No evidence was found of any mutations in any copy of *serTGA* or its promoter; all 245 bp were covered by a minimum of 127 reads in each duplication strain, with at least 94% of reads at each position carrying the dominant base.

• Supplementary file 6. Reference list of 42 unique tRNA sequences in *P. fluorescens* SBW25 GtRNAdb 2.0 predicts 67 tRNA genes in *P. fluorescens* SBW25 (*Chan and Lowe, 2016*). These include 42 unique primary tRNA sequences, each of which is listed in this file. These 42 sequences are used as references to align the YAMAT-seq data in this work (see *Supplementary file 7*). Note that the list of 42 sequences includes one likely pseudo tRNA (11_Cys-GCA-2–1). This sequence is not predicted to form a tRNA with conserved cloverleaf secondary structure (*Chan and Lowe, 2019*). Further, no YAMAT-seq sequences were aligned to this reference sequence in any sample. We conclude that Cys-GCA-2–1 does not form part of the functional mature tRNA pool in SBW25. In addition, the *serCGA* sequence (30_Ser-CGA-1–1) is expected to be absent from 21 of 27 samples (samples 2–8, 11–17, and 20–26); this tRNA is encoded by *serCGA* (the gene that was deleted by genetic engineering in this work and remains absent in all derived strains). As expected, almost no reads were obtained for this reference sequence in these 21 samples. The very low numbers of Ser-CGA reads obtained in some of these samples (e.g., two reads in sample 20, the third replicate of Δ*serCGA*-1) are likely to be barcode misallocations from one of the six SBW25 or W1-L samples.

• Supplementary file 7. YAMAT-seq data showing elimination of tRNA-Ser(CGA) followed by elevating of tRNA-Ser(UGA) expression in each of the five duplication isolates. The first tab contains index details and a summary of the raw YAMAT-seq reads (GEO accession number GSE144791; *Edgar et al., 2002*) for each of the 27 samples (three replicates of nine strains): a minimum of 636,995 reads of the expected size (80–151 bp) was obtained *per* sample. In each case, between 86.4% and 93.0% of these aligned to the list of 42x reference SBW25 tRNA sequences (provided in *Supplementary file 6*). The subsequent nine tabs contain the YAMAT-seq data for each of the nine strains tested (SBW25, Δ*serCGA*-1, Δ*serCGA*-2, W1-L, M1-L, M2-L, M2-Lop, M3-L, and M4-L). Each tab contains (i) the numbers of reads for 42 reference tRNAs, for three replicates (left), (ii) numbers of reference reads for 39 tRNA types in SBW25 (e.g., tRNA-Asn-GTT is the sum of reference sequences 7_Asn-GTT-1–1 and 8_Asn-GTT-2–1; middle), (iii) the proportion of each tRNA type in the mature tRNA pool for each of the three samples (right), and (iv) a scatter plot of the YAMAT-seq proportions versus the proportion of the tRNA gene set encoding the tRNA type. Blue = tRNA-Ser(CGA), green = tRNA-Ser(UGA). The final tab contains information regarding the unused reads for each sample.

• Transparent reporting form

### Data availability
Illumina whole genome sequencing data has been uploaded to NCBI SRA (accession PRJNA558233). YAMAT-seq data has been uploaded to NCBI GEO (accession GSE144791). Source data files have been provided for Figures 2B, 2C, 2D, 2F, 2G, 2H, 3B, 3C, 3D, 3E, 4C, 4D, 5A, 5B and Figure 5—figure supplement 1.

The following datasets were generated:

| Author(s) | Year | Dataset title | Dataset URL | Database and Identifier |
|---|---|---|---|---|
| Gallie J | 2019 | Experimental evolution of a bacterial strain with a sub-optimal tRNA gene set (single-copy tRNA gene serCGA deleted) | https://www.ncbi.nlm.nih.gov/bioproject/PRJNA558233/ | NCBI BioProject, PRJNA558233 |
| Gallie J, Ayan GkeB, Park HJ | 2019 | YAMAT-seq of mature tRNA pools in the bacterium Pseudomonas fluorescens SBW25 and derivatives | https://www.ncbi.nlm.nih.gov/geo/query/acc.cgi?acc=GSE144791 | NCBI Gene Expression Omnibus, GSE144791 |

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
