## [Decision Letter]

**Acceptance summary:**

This study shows that deletion of a non-essential single-copy tRNA gene in *Pseudomonas* alters the cellular tRNA pool and reduces fitness, especially when conditions enable rapid growth. During experimental evolution in the laboratory, they find that the tRNA deletion can be compensated by repeated, large duplications of a part of the genome, which include a near cognate tRNA gene. This work demonstrates effects of tRNA gene redundancy on fitness and the means by which genomes can rapidly compensate for the loss of redundancy.

**Decision letter after peer review:**

Thank you for submitting your article "The birth of a bacterial tRNA gene" for consideration by *eLife*. Your article has been reviewed by three peer reviewers, and the evaluation has been overseen by a Reviewing Editor and Patricia Wittkopp as the Senior Editor. The reviewers have opted to remain anonymous.

The reviewers have discussed the reviews with one another and the Reviewing Editor has drafted this decision to help you prepare a revised submission.

Summary:

This well-written manuscript demonstrates that deletion of a non-essential single-copy tRNA gene in *Pseudomonas* fluorescens (ser-tRNA CGA) alters the cellular tRNA pool and reduces fitness. During experimental evolution in the laboratory, the authors find that the tRNA deletion is compensated by repeated, large duplications of a part of the genome, which include a near cognate tRNA gene (tRNA TGA). The duplications are associated with increased tRNA TGA expression and increased fitness. The authors suggest that this is a novel evolutionary response to overcome translational inefficiency. These results are framed by a simple model of translation dynamics to understand the initial fitness effects of the gene deletion, and the observed evolutionary response. Overall, the reported experimental work is well done and presents interesting results. The manuscript is also clearly written. However, all reviewers raised questions about the presented mathematical/verbal model and agree that the novelty and breadth of the findings are overstated. Evolution via gene duplication is a well-known phenomenon in evolutionary biology, has been observed in many experimental evolution studies, and is the most likely outcome of the experimental design. The repeated observation of duplications of a region containing tRNA-TGA as well as other tRNAs is a worthwhile finding but the generality of this result for our broader understanding of the evolution of tRNAs requires further exploration.

Essential revisions:

1) Please consider more of the literature on tRNA pool evolution and clarify how this study represents a significant advance. There has been much discussion of gene loss and duplication as key features (e.g. Withers et al., 2006; Wald and Margalit 2014; Tremblay-Savard et al., 2015), and the resulting evolutionary flexibility of tRNA gene sets (e.g. Ikemura, 1985; Rocha, 2004; Higgs and Ran, 2008; Diwan et al., 2018). This paper provides experimental support for these ideas, arising from the deletion of a single bacterial tRNA gene. This is a valuable result, being the first such demonstration in bacteria. However (contrary to the projection in the manuscript), this is not an unexpected result, and is not sufficient to generalize broadly. The reported adaptation of YAMAT-seq to measure bacterial tRNAs is very useful. Prior models of tRNA gene set evolution have demonstrated the importance of codon usage bias for translation rate (e.g. Bulmer, 1991; Berg et al., 1997; Higgs and Ran, 2008).

2) The initial model is overly simplistic and ignores much of the advances made in our understanding from prior work showing the importance of codon usage (see references above). The model does not explicitly include links between tRNA set and translation rate, and between translation rate and fitness; these are instead left as verbal arguments. Equation 1 is odd, because it suggests that codon B is translated by both alpha and beta (whereas only one tRNA can decode a codon at a time) and perhaps only works when alpha is limiting. The meaning of Equation 3 is unclear. The calculated translation times (Equation 4) should probably be clearly discussed as relative (not absolute) times. Finally, none of the model predictions are novel (see references above). Citing and discussing prior work may be sufficient to clearly set up the basic premise here (instead of the model), allowing a deeper focus on the experimental work.

3) We are not convinced that a key assumption of the model is reasonable: " the rate of translation by an anticodon-codon pair is determined solely by the proportion of the anticodon in the tRNA pool". I would think that the rate-limiting step in the translation of a codon is determined by the stochastic search for the cognate ternary complex (aminoacyl-tRNA+EF-Tu and GTP) to the A-site (Varenne et al., 1984). I cannot see that this is directly related to the proportion in the tRNA pool, but mainly to the concentration of each cognate ternary complex at steady-state. Reducing the concentration drastically by deleting a tRNA gene is likely to be limiting for growth, but this can be compensated for by increasing the concentration by a duplication. If the authors assume that competition with non-cognate and near-cognate ternary complexes are of major importance for the rate of translation of a codon this should be clearly stated. The authors find that tRNA proportions vary almost 100,000-fold; does this mean that translation rates are also expected to vary 100,000-fold and is there any experimental support for this? Please provide a clear explanation for why proportion of the anticodon in the tRNA pool is expected to be rate-limiting, supported by proper references.

4) A number of factors relevant to understand translation rates and tRNA gene evolution are not discussed in sufficient depth, and as early in the manuscript as is necessary. For instance, codon usage doesn't feature in the Results section until much later. So the experimental results seem puzzling until it is clear that the non-essential tRNA gene actually recognizes a codon that is very abundant in the genome. In the Discussion section (subsection “Retention of *serCGA* in *P. fluorescens* SBW25 wild type”), selection due to codon bias should be considered as a 4th hypothesis for the retention of tRNA(CGA) (perhaps in combination with hypotheses 1 or 3). Prior work also shows that tRNA modifications can alter the accuracy and efficiency of translation (Grosjean et al., 2010; Bjork and Hagervall, 2014; Manickam et al., 2015). These details are mentioned in passing, but deserve more prominence because they are really critical to set expectations and interpret the results. The focal tRNA species are expected to be modified by cmoAB; if the *Pseudomonas* strain used here has this modification system, it could explain the observed results: modified tRNA(UGA) can compensate tRNA(CGA) function, whereas the other near cognate tRNAs cannot (it is unclear whether G-U wobble works when G is in the codon and U is in the anticodon). I suggest that the details about codon usage, gene copy numbers of all cognate and near-cognate tRNAs, and relevant modification systems should be presented and clearly discussed at the outset.

5) We agree that the increase in tRNA(UGA) levels probably drove the large duplications observed during evolution. However, there are some points of concern here.

a) Given that the deletions were large, it would be useful to be able to estimate the contribution of tRNA(UGA) to increased fitness. Does deleting the duplicated tRNA(UGA) in evolved isolates reduce fitness, and by how much? Related to this, it was not clear whether there were any other mutations in the evolved lines, and their identity; e.g. were there any promoter mutations in the native copy of the tRNA(UGA) gene?

b) What is the level of overexpression of the tRNA gene on the plasmid (Figure 4)? If this is much more than the 2-fold increase due to gene duplication, it means that we do not know if a 2-fold increase is sufficient to increase fitness. On a related note, I could not see information on the sensitivity of the tRNA-seq method (I might have missed this); this is necessary to know how much confidence to place in the measured fold change values.

[Editors' note: further revisions were suggested prior to acceptance, as described below.]

Thank you for submitting your article "The birth of a bacterial tRNA gene by large-scale, tandem duplication events" for consideration by *eLife*. Your article has been reviewed by two peer reviewers, and the evaluation has been overseen by a Reviewing Editor and Patricia Wittkopp as the Senior Editor. The reviewers have opted to remain anonymous.

The reviewers have discussed the reviews with one another and the Reviewing Editor has drafted this decision to help you prepare a revised submission.

Summary:

The reviewers are mostly satisfied with the response to the prior set of reviews and appreciate the well-written presentation. A few points remain to be addressed to clarify assumptions and discuss caveats of your conclusions.

Revisions:

1) Please specify your assumption that all tRNAs are fully charged, maybe with reference that this is not always the case and description of what YAMAT-Seq measures. We are not quite satisfied with the response to Essential revision point 3 where the authors were asked to explain and justify the assumptions made to be able to say that the translation rate of a codon is determined solely by the proportion of anticodon in the tRNA pool. In the revised version the envisaged translation system is explained more clearly and key assumptions, such that EF-Tu binds all types of tRNA with equal affinity, are explicitly stated. However, the process of charging of tRNAs by aminoacyl-tRNA synthetases/tRNA-ligases and assumptions about charging levels of tRNAs remains incompletely considered. The authors only refer to mature tRNAs in the text, but I am not sure if this includes both charged (aa-tRNA) and uncharged tRNAs. This leads to a number of questions about experimental data, assumptions and what is included in the model:

a) Does YAMAT-seq measure both charged and uncharged tRNAs?

b) Do you assume that all tRNAs are fully charged? This might be reasonable in minimal media (for example Kimberly A Dittmar, Michael A Sørensen, Johan Elf, Måns Ehrenberg, Tao Pan. Selective charging of tRNA isoacceptors induced by amino-acid starvation. EMBO Rep 2005 Feb;6(2):151-7 and references therein).

The focus on serine is potentially problematic. Serine is one of the most toxic amino acids and it is possible that the reduction in growth rate in the deletion mutants are mainly due to this toxicity rather than a reduced translation rate, which would provide an alternative explanation for why the effect on growth is much smaller in minimal media. The proteose peptone 3 used in KB medium is high in serine (about 12% of total amino acids) suggesting that this might cause toxicity due to the inability of L-serine-deaminase to degrade excess serine after a reduction in ser-tRNA concentration. Addressing this in the final manuscript would make readers aware of this issue, even while pointing out that this explanation may be insufficient because adding a plasmid with the amplified tRNA increases fitness.

2) Related to this point, in rich media it has been seen that charging levels for serine tRNAs can be very low at below 10% although serine concentration is high (Avcilar-Kucukgoze et al., 2016). This work is new to this reviewer, but this might be of interest to the authors and readers of the article.

---

## [Author Response]

Essential revisions:1) Please consider more of the literature on tRNA pool evolution and clarify how this study represents a significant advance. There has been much discussion of gene loss and duplication as key features (e.g. Withers et al., 2006; Wald and Margalit, 2014; Tremblay-Savard et al., 2015), and the resulting evolutionary flexibility of tRNA gene sets (e.g. Ikemura, 1985; Rocha, 2004; Higgs and Ran, 2008; Diwan et al., 2018). This paper provides experimental support for these ideas, arising from the deletion of a single bacterial tRNA gene. This is a valuable result, being the first such demonstration in bacteria. However (contrary to the projection in the manuscript), this is not an unexpected result, and is not sufficient to generalize broadly. The reported adaptation of YAMAT-seq to measure bacterial tRNAs is very useful. Prior models of tRNA gene set evolution have demonstrated the importance of codon usage bias for translation rate (e.g. Bulmer, 1991; Berg et al., 1997; Higgs and Ran, 2008).

To address this comment, we have re-written substantial portions of the manuscript, including the Abstract, Introduction, and Discussion. Specifically:

1) The Introduction has been re-written to include a more comprehensive review of the literature on tRNA gene set evolution. It now includes a deeper analysis of factors that influence tRNA gene set evolution (including post-transcriptional modification pathways, and codon use; and the mechanisms of evolutionary change. The suggested references have been included, and we thank the reviewers for these useful suggestions.

2) The Abstract, final paragraph of the Introduction, and the final paragraph of the Discussion have been re-written to clarify the advances made in our study. In our opinion, the main advance provided by our work is the direct observation of the evolution of a tRNA gene set by the mechanism of gene duplication (Introduction). In this way, our work provides empirical support for a number of phylogenetic and computational studies that postulate gene duplication as a mechanism of bacterial tRNA gene set evolution (Tremblay-Savard et al., 2015; Wald and Margalit, 2014; Withers et al., 2006). We agree that our results alone are not sufficient to generalize about the prevalence of duplication events in tRNA gene set evolution, and have included this in our Discussion. A second advance provided by our work is the adaptation of YAMAT-seq for use in bacteria (Introduction). We are pleased that the reviewers think this is useful, and hope that it will aid others in their research.

2) The initial model is overly simplistic and ignores much of the advances made in our understanding from prior work showing the importance of codon usage (see references above). The model does not explicitly include links between tRNA set and translation rate, and between translation rate and fitness; these are instead left as verbal arguments. Equation 1 is odd, because it suggests that codon B is translated by both alpha and beta (whereas only one tRNA can decode a codon at a time) and perhaps only works when alpha is limiting. The meaning of Equation 3 is unclear. The calculated translation times (Equation 4) should probably be clearly discussed as relative (not absolute) times. Finally, none of the model predictions are novel (see references above). Citing and discussing prior work may be sufficient to clearly set up the basic premise here (instead of the model), allowing a deeper focus on the experimental work.

The response to Essential revision point 3 (below) also contains information relevant to this point.

From the reviewers’ comments, we can see that our explanation of the initial model was unclear. For example, we did not intend to imply that a single codon is translated by more than one tRNA molecule at any given time; rather, in cases where a codon can be translated by more than one type of tRNA, the proportions of these tRNAs were added together to estimate the average time required to randomly sample a matching tRNA from the available pool (see also the summary provided in response to Essential revision 3). The initial model was indeed simple. It was intended as a null model demonstrating one simple point: translational speed is optimized by eliminating tRNA redundancy (in the absence of complicating factors). Hence, in cases where surplus tRNAs are retained (such as *serCGA* in *P. fluorescens* SBW25), additional factors should be considered. An example factor – different translational efficiencies between codon-tRNA pairings – was provided by the extended model.

However, on reflection, we agree with the reviewers that the manuscript would benefit from reorganization to allow more focus on the main novelty of our study: the experimental work. As suggested, we have removed the initial modelling section, instead discussing more prior work in the Introduction (including work on the effects of codon use, links between tRNA set and translation rate, and links between translation rate and growth/fitness). The results now begin with a new section, “The *P. fluorescens* SBW25 tRNA gene set”. This section covers the structure of the tRNA gene set, with emphasis on the seryl-tRNAs and codons that feature later in our work. A (re-written) modelling section appears at the end of the Results section, with the aim of providing a cohesive molecular explanation of the observed results.

3) We are not convinced that a key assumption of the model is reasonable: " the rate of translation by an anticodon-codon pair is determined solely by the proportion of the anticodon in the tRNA pool". I would think that the rate-limiting step in the translation of a codon is determined by the stochastic search for the cognate ternary complex (aminoacyl-tRNA+EF-Tu and GTP) to the A-site (Varenne et al., 1984). I cannot see that this is directly related to the proportion in the tRNA pool, but mainly to the concentration of each cognate ternary complex at steady-state. Reducing the concentration drastically by deleting a tRNA gene is likely to be limiting for growth, but this can be compensated for by increasing the concentration by a duplication. If the authors assume that competition with non-cognate and near-cognate ternary complexes are of major importance for the rate of translation of a codon this should be clearly stated. The authors find that tRNA proportions vary almost 100,000-fold; does this mean that translation rates are also expected to vary 100,000-fold and is there any experimental support for this? Please provide a clear explanation for why proportion of the anticodon in the tRNA pool is expected to be rate-limiting, supported by proper references.

These comments, together with those in Essential revision point 2, lead us to conclude that there is a miscommunication about how we envisage the translational system and effects of the observed mutations. To clarify our position we have written the following summary (including definitions, citations, and details of where the relevant information can now be found in the manuscript):

Summary

Ternary complexes versus the mature tRNA pool.

Prokaryotic translation consists of three main stages: initiation, elongation, and termination (reviewed in Rodnina, 2018). The rate limiting step of elongation is the stochastic search for a ternary complex to match the codon occupying the ribosomal A site (Varenne et al., 1984). Ternary complexes consist of a mature tRNA, elongation factor EF-Tu, and GTP (Bensch et al., 1991). EF-Tu binds all types of mature tRNA with approximately equal affinity (Louie et al., 1984; Ott et al., 1990). Hence the composition of the ternary complex pool is expected to reflect that of the mature tRNA pool; proportions of mature tRNAs provide a barometer for ternary complex availability, and ultimately elongation speed. This information is now included in the Introduction and Results.

Competition between cognate and near-cognate tRNAs.

A codon-tRNA match is considered “cognate” when the first two bases of the codon form Watson-Crick base pairs with the tRNA anticodon *and* the third codon base forms either a Watson-Crick or wobble base pair (Plant et al., 2007). Hence, in our system, both tRNA-Ser(CGA) and tRNA-Ser(UGA) are considered to be cognate matches for codon UCG (and tRNA-Ser(UGA) is also a cognate match for codons UCA and UCU). This means that, while a single UCG codon is translated by a single tRNA molecule, stochastic sampling of either a tRNA-Ser(CGA) or tRNA-Ser(UGA) ternary complex is expected to result in translation. This is why the proportions of tRNA-Ser(CGA) and tRNA-Ser(UGA) are added together to estimate the average time required to match a UCG codon (Equation 2B) (this point may also help to clarify some parts of Essential revision point 2).

For simplicity, our model assumes that the two tRNA types translate UCG with equal efficiency. The possibility that UCG is translated more efficiently by tRNA-Ser(CGA) than tRNA-Ser(UGA) provides one reason why duplication of *serTGA* does not fully restore wild type fitness, and may contribute to the retention of *serCGA* in the wild type strain (as opposed to 2+ copies of *serTGA*) (Discussion). We have not referred to tRNA-Set(CGA) and tRNA-Ser(UGA) as being “in competition” for translating UCG because, in our view, they are better described as synergistic: the translational demand imposed by UCG codons is, presumably, shared between the two types. We have included a new section at the beginning of the Results (“*The* P. fluorescens *SBW25 tRNA gene set*”) in which we outline the codons, tRNAs, and matching patterns that are important for understanding the manuscript. This new section, together with the extended Introduction, aims to clarify these issues from the outset.

The effects of our mutations on translation.

As outlined above, UCG codons are expected to be translated by tRNA-Ser(CGA) or tRNA-Ser(UGA) in wild type *P. fluorescens* SBW25. Deletion of *serCGA* eliminates tRNA-Ser(CGA), presumably resulting in all UCG codons being translated by tRNA-Ser(UGA). Hence, in a *serCGA* deletion mutant, a lower proportion of ternary complexes match UCG, leading to an increase in the average time needed to stochastically sample a matching ternary complex (and ultimately slowing translation/growth). As the reviewers correctly surmise, the translational and selective pressure exerted on tRNA-Ser(UGA) by *serCGA* deletion can be relieved by elevating tRNA-Ser(UGA) levels, through increasing *serTGA* gene copy number. We have included this explanation in the final Results section and Discussion.

We appreciate that the original manuscript did not sufficiently lay out the relationship between UCG codons and tRNAs. We hope that the new organization and sections of the manuscript clarify these issues, and that the results of our experiments are now more intuitive.

The reviewers bring up some very good points regarding the YAMAT-seq measures. As they point out, the within-strain mature tRNA proportions measured by YAMAT-seq vary about 100,000 fold. Even after removing the three lowest tRNA proportions (which are thought to be affected by post-transcriptional modifications impeding the reverse transcription reaction), the variation in tRNA proportions exceeds reported variations in codon translation rates (Gardin et al., 2014). This highlights that, while the YAMAT-seq results provide a useful overview of the relative amounts of tRNAs in a mature tRNA pool, within-strain comparisons should be treated with caution. The real strength of the YAMAT-seq data lies in detecting differences in tRNA proportions across samples. In our case, this means detected the effects of *serCGA* deletion and subsequent *serTGA* duplication on the mature tRNA pool of SBW25. Hence, as the reviewers also point out in Essential revision point 2, interpretations of the YAMAT-seq data should focus on relative proportions (and, hence, relative translation times) across strains. We have altered the YAMAT-seq results and model section to reflect these comments.

4) A number of factors relevant to understand translation rates and tRNA gene evolution are not discussed in sufficient depth, and as early in the manuscript as is necessary. For instance, codon usage doesn't feature in the Results section until much later. So the experimental results seem puzzling until it is clear that the non-essential tRNA gene actually recognizes a codon that is very abundant in the genome. In the Discussion section (subsection “Retention of serCGA in P. fluorescens SBW25 wild type”), selection due to codon bias should be considered as a 4th hypothesis for the retention of tRNA(CGA) (perhaps in combination with hypotheses 1 or 3). Prior work also shows that tRNA modifications can alter the accuracy and efficiency of translation (Grosjean et al., 2010; Bjork and Hagervall, 2014; Manickam et al., 2015). These details are mentioned in passing, but deserve more prominence because they are really critical to set expectations and interpret the results. The focal tRNA species are expected to be modified by cmoAB; if the Pseudomonas strain used here has this modification system, it could explain the observed results: modified tRNA(UGA) can compensate tRNA(CGA) function, whereas the other near cognate tRNAs cannot (it is unclear whether G-U wobble works when G is in the codon and U is in the anticodon). I suggest that the details about codon usage, gene copy numbers of all cognate and near-cognate tRNAs, and relevant modification systems should be presented and clearly discussed at the outset.

We agree with the reviewers, and have changed the layout of the manuscript to ensure early and clear introduction of the suggest concepts and their relevance to our experimental system. Specifically:

1) We have outlined the role of codon-tRNA matching patterns (and, hence, post-transcriptional modification enzymes) and codon use in the Introduction.

2) We have written a new first Results section (“*The* P. fluorescens *SBW25 tRNA gene set*”) outlining the key elements of our experimental system, and the relationships between them (*e.g.*, seryl tRNA genes, serine codon use, codon matching patterns). We would like to thank the reviewers for the particularly helpful suggestions regarding the Cmo-mediated modification of tRNA-Ser(UGA), of which we were not previously aware.

5) We agree that the increase in tRNA(UGA) levels probably drove the large duplications observed during evolution. However, there are some points of concern here.a) Given that the deletions were large, it would be useful to be able to estimate the contribution of tRNA(UGA) to increased fitness. Does deleting the duplicated tRNA(UGA) in evolved isolates reduce fitness, and by how much? Related to this, it was not clear whether there were any other mutations in the evolved lines, and their identity; e.g. were there any promoter mutations in the native copy of the tRNA(UGA) gene?

While we agree that deleting one *serTGA* copy would be a nice addition to our experiments, this particular genetic manipulation is unlikely to be successful. This is largely due to the instability of large-scale duplication fragments; similar fragments are lost at rates of 1 % in overnight culture (Anderson and Roth, 1977; Reams and Roth, 2015), and ongoing investigations indicate that our duplication fragments are no exception (unpublished data). Duplication fragment loss typically occurs via RecA-mediated, homologous recombination (Reams and Roth, 2015) – the same process that would hypothetically be harnessed during the genetic engineering process to replace a *serTGA* copy with a ~1 kb deletion fragment. Successfully targeting the ~100 bp *serTGA* without affecting the remainder of the ~45 kb duplication fragment is highly unlikely. Even in the event of successful manipulation, the desired engineered strain would be expected to quickly lose the remainder of the duplication fragment during growth (given that loss of duplication fragments occurs at high rates, and no selective advantage is expected in the absence of the second *serTGA* copy).

With the above in mind, we concentrated our resources on demonstrating that an increase in *serTGA* expression can indeed compensate for *serCGA* loss. Expression of *serTGA* from the pSXn plasmid improves growth of the *serCGA* deletion mutant and, importantly, does not affect the growth of the wild type strain (Figure 3C). This result is central to our work, because it provides empirical evidence that codon UCG actually can be translated by tRNA-Ser(UGA) (Figure 1B), and hence can compensate for tRNA-Ser(CGA) loss. However, we do agree with the reviewers that other genes within the 45 kb duplication fragment could still contribute via unknown mechanisms to the compensatory effect. To reflect this, we have amended the relevant Results section: “…the result that pSXn-based *serTGA* expression specifically improves the growth rate of ∆*serCGA* demonstrates that *serTGA* can provide a degree of compensation for *serCGA* loss. Other genes in the shared 45 kb fragment may nevertheless contribute to compensation, via unidentified mechanisms.”.

Regarding mutations other than the large-scale duplications: only one additional putative mutation was identified among the mutant isolates: M2-L contains a synonymous mutation in carbohydrate metabolism gene *edd* (in addition to carrying a 192 kb duplication fragment). This mutation is not expected to have major effects on our translational system because (i) it is a synonymous change to an arginine (*i.e.* not a serine) codon, resulting in no change to the Edd protein sequence, and (ii) it is not found in any other isolate, including M2-Lop that was isolated from the same population at the same time point. This information is now explicitly included in the Results. A number of intergenic differences in homopolymeric tracts and other repeats were called in all/most isolates (mutant and wild type); these have also been previously called in other, unrelated evolution experiments (*i.e.*, they are either mistakes in the published SBW25 genome sequence, or are present in our stored wild type strain) (Beaumont et al., 2009; Gallie et al., 2019). A comprehensive list of all differences called by *breseq* (Deatherage and Barrick, 2014) and Geneious (v11.1.4), and accompanying descriptions, is in Supplementary file 3.

b) What is the level of overexpression of the tRNA gene on the plasmid (Figure 4)? If this is much more than the 2-fold increase due to gene duplication, it means that we do not know if a 2-fold increase is sufficient to increase fitness. On a related note, I could not see information on the sensitivity of the tRNA-seq method (I might have missed this); this is necessary to know how much confidence to place in the measured fold change values.

Regarding the pSXn plasmid: we do not know the exact level of expression from the pSXn plasmid. We can say that the inserted tRNA gene is placed under IPTG-inducible expression from a strong (*tac*) promoter (de Boer et al., 1983). We do not add IPTG at any stage of our experiments, meaning that we expect only low level, leaky expression from the repressed promoter (Owen and Ackerley, 2011). However, as the reviewers point out, this expression is likely to be higher than a two-fold increase. We have made this explicit in the interpretation of our results: “While it should be noted that tRNA-Ser(UGA) levels resulting from pSXn-based expression are likely to exceed those resulting from an additional chromosomal copy of *serTGA*, the result that pSXn-based *serTGA* expression specifically improves the growth rate of ∆*serCGA* demonstrates that *serTGA* can provide a degree of compensation for *serCGA* loss.”. We have also included more details on the plasmid, and construction process (new Materials and methods section “Expression of tRNA genes from the pSXn plasmid”).

To assist with interpreting the strength of the YAMAT-seq assay, we have provided more details (*e.g.*, raw read numbers, the numbers of reads aligning to each tRNA types) within the main text (Results) and Materials and methods. Here, we would like to highlight the considerable weight of the YAMAT-seq data. Firstly, this is a large data set that is consistent across replicates; each of the nine strains were assayed in three biological replicates, and no major differences were seen between tRNA proportions in among replicates (Supplementary file 7). Secondly, a robust statistical analysis that corrects for multiple comparisons was employed to detect differences in relative tRNA expression levels between strains (DESeq2; Anders and Huber, 2010; Love et al., 2014). Importantly, DESeq2 control comparisons revealed no significant differences in tRNA expression levels (*i.e.*, no differences were detected between SBW25 and the Day 13 wild type lineage isolate, or between the two biological replicates of the *serCGA* deletion strain; Figures 5A, 5B). Thirdly, the differences detected were consistent: *serCGA* deletion results in (permanent) loss of tRNA-Ser(CGA), and a rise in tRNA-Ser(UGA) was the only consistently significant difference upon duplication of *serTGA*. Overall, we have confidence in the differences detected in tRNA expression levels between strains. To what extent smaller changes are not detected as significant, we cannot currently say.

[Editors' note: further revisions were suggested prior to acceptance, as described below.]

Revisions:1) Please specify your assumption that all tRNAs are fully charged, maybe with reference that this is not always the case and description of what YAMAT-Seq measures. We are not quite satisfied with the response to Essential revision point 3 where the authors were asked to explain and justify the assumptions made to be able to say that the translation rate of a codon is determined solely by the proportion of anticodon in the tRNA pool. In the revised version the envisaged translation system is explained more clearly and key assumptions, such that EF-Tu binds all types of tRNA with equal affinity, are explicitly stated. However, the process of charging of tRNAs by aminoacyl-tRNA synthetases/tRNA-ligases and assumptions about charging levels of tRNAs remains incompletely considered. The authors only refer to mature tRNAs in the text, but I am not sure if this includes both charged (aa-tRNA) and uncharged tRNAs. This leads to a number of questions about experimental data, assumptions and what is included in the model:a) Does YAMAT-seq measure both charged and uncharged tRNAs?

The YAMAT-seq procedure measures both charged and uncharged mature tRNAs. Specifically, the YAMAT-seq procedure is as follows (Shigematsu et al., 2017; Warren et al., 2020):

1) Total RNA is isolated from growing cells. This includes mRNA, rRNA, and small RNAs such as tRNAs (including pre-tRNAs, mature uncharged tRNAs, and mature charged tRNAs…).

2) Total RNA is treated with Tris-HCl (pH 9.0), removing the amino acids from charged tRNAs. The end result is that (most) mature tRNAs are now uncharged.

3) Y-shaped, DNA/RNA hybrid adapters (Author response image 1) and T4 RNA ligase 2 are added. The adapters ligate specifically to nucleic acid molecules with a single-stranded, exposed 5'‑NCCA‑3' end in close proximity with a 3'‑inorganic phosphate-5' (*i.e.*, mature, uncharged tRNAs) (Author response image 1). They do not bind efficiently to (for example) mature charged tRNAs, pre-tRNAs, tRNAs with damaged 5'‑NCCA‑3' or 3'‑Pi‑5' ends, or tRNA fragments.

4) Adapter-tRNA complexes are reverse transcribed (RT primer; Illumina), products PCR-amplified (F and R primers; Illumina) and purified from a native polyacrylamide gel.

5) The pool of DNA molecules is deep sequenced with Illumina.

**Author response image 1. sa2fig1:** YAMAT-seq (Shigematsu et al., 2017) quantifies mature uncharged and charged tRNAs in growing cells. Total RNA isolated for YAMAT-seq includes both charged and uncharged tRNAs. The amino acids are removed from charged tRNAs in alkaline conditions. Y-shaped, DNA/RNA hybrid adapters (**A**) are ligated specifically to mature, uncharged tRNAs (**B**). Adapter-tRNA complexes are reverse transcribed and products PCR amplified, purified, and deep sequenced (Illumina). Pi=inorganic phosphate, RT=reverse transcription (RT) primer, F= forward PCR primer, R=reverse PCR primer.

To clarify the above, a specific statement about charged and uncharged mature tRNAs has been included in the YAMAT-seq results, and some details have been added to the YAMAT-seq method section.

b) Do you assume that all tRNAs are fully charged? This might be reasonable in minimal media (for example Kimberly A Dittmar, Michael A Sørensen, Johan Elf, Måns Ehrenberg, Tao Pan. Selective charging of tRNA isoacceptors induced by amino-acid starvation. EMBO Rep 2005 Feb;6(2):151-7 and references therein).

The reviewers make a very good point; our model uses our YAMAT-seq data – which measures both charged and uncharged mature tRNAs (see **point 1a** above) – while assuming that all mature tRNAs are charged.

Charged versus uncharged mature tRNA proportions have been systematically measured for all *E. coli* tRNA species in two different studies (Avcilar-Kucukgoze et al., 2016; Dittmar et al., 2005). These studies collectively report that in nutrient poor media, all tRNA species are fairly uniformly charged at ~60-90 %. However, during exponential growth in rich medium (LB), charging levels are reported to vary considerably between tRNA species (Avcilar-Kucukgoze et al., 2016). In particular, the four seryl-tRNAs were reported to show very low charging levels (~10 %) under these conditions (see their Figure 1A).

We find this result somewhat surprising. Uncharged tRNAs interact with RelA, a ribosome-associated protein that brings uncharged tRNAs to the ribosomal A site (Winther et al., 2018). In this respect, uncharged tRNAs compete with charged tRNAs (in the form of ternary complexes) for occupation of the ribosomal A site; uncharged tRNAs slow elongation or, at higher levels, activate the stringent response (Goldman and Jakubowski, 1990). The stringent response involves a global shift in gene expression, ultimately resulting in slower growth and division (reviewed in Hauryliuk et al., 2015). In light of the need for rapid growth and division, it seems counterintuitive that uncharged tRNAs would exist in high proportions during exponential growth in rich medium. While Avcilar-Kucukgoze et al. do not discuss the low seryl-tRNA charging levels with respect to the stringent response, we note that *E. coli* MC4100 – the strain in which the majority of their measurements were made – carries the *relA1* mutation, which effectively eliminates uncharged tRNA-based activation of the strignent response (Metzger et al., 1989).

Clearly, a charging rate of only 10 % would violate our assumption that all mature seryl-tRNAs are charged. Reassuringly however, under rapid growth conditions all within-family tRNA types (*i.e.*, those carrying the same amino acid) show similar charging levels under all conditions measured so far (Avcilar-Kucukgoze et al., 2016; Dittmar et al., 2005). If similar, consistently low charging levels exist for SBW25 seryl-tRNAs during YAMAT-seq, our general conclusions are expected to hold given that the model uses only seryl-tRNA proportions (which, if flawed, are presumably consistently so) to estimate elongation times.

We have added an explicit statement of our assumption and its limitations that all tRNAs are charged in the model section of the Results, and outlined why this is a limitation of the model.

The focus on serine is potentially problematic. Serine is one of the most toxic amino acids and it is possible that the reduction in growth rate in the deletion mutants are mainly due to this toxicity rather than a reduced translation rate, which would provide an alternative explanation for why the effect on growth is much smaller in minimal media. The proteose peptone 3 used in KB medium is high in serine (about 12% of total amino acids) suggesting that this might cause toxicity due to the inability of L-serine-deaminase to degrade excess serine after a reduction in ser-tRNA concentration. Addressing this in the final manuscript would make readers aware of this issue, even while pointing out that this explanation may be insufficient because adding a plasmid with the amplified tRNA increases fitness.

If we have understood correctly, the reviewers are suggesting that *serCGA* deletion may cause an increase in free intracellular serine, the toxicity of which could become problematic in the presence of high amounts of extracellular serine. This could provide an alternative explanation for why the ∆*serCGA* growth defect is seen in KB (serine rich) and not M9 (serine poor) medium. The toxicity of serine is an aspect of our work that we had not fully considered, and we thank the reviewers for the suggestion.

On consideration, we think that while serine toxicity may contribute to the growth defect in KB medium, it is unlikely to account for the entire effect. This is due to three reasons:

1) Excess serine is highly toxic and thus tightly regulated (Avcilar-Kucukgoze et al., 2016; Kriner and Subramaniam, 2019). Studies of the three L-serine deaminases in *E. coli* show Michaelis constants (Km) in the millimolar range (Burman et al., 2004; Cicchillo et al., 2004), meaning that these L-serine deaminases have a high affinity for L-serine. This is consistent with L-serine deaminases playing a role in preventing serine excess (Zhang and Newman, 2008). Indeed, serine is depleted from rich medium much more rapidly than any other amino acid during bacterial growth (Zhang et al., 2010).

Like *E. coli*, *P. fluorescens* SBW25 is predicted to encode three L-serine deaminases (*sdaA1/pflu1035, tdcG/pflu4898, sdaA2/pflu5679*), and a transmembrane L-serine transporter (*sdaC/pflu1034*). Hence, SBW25 seems adequately equipped to deal with the (presumably relatively small) amounts of excess serine that may be produced upon the elimination of tRNA-Ser(CGA), a tRNA that accounts for ~1.5 % of the SBW25 tRNA pool (see Supplementary file 7). Notably, none of the L-serine deaminases or the serine transporter are found within any of the duplication fragments (see Supplementary file 4).

2) While serine is indeed toxic for *E. coli*, it is possible to delete all three genes encoding L-serine deaminases and grow the triple mutant in serine rich medium (Zhang and Newman, 2008). Cells of the triple mutant grown in this manner are deformed during the early stages of growth, indicating effects of serine accumulation on cell growth and division (Zhang et al., 2010; Zhang and Newman, 2008). The misshapen cells have been proposed to result from problematic cell wall synthesis, possibly through the mis-incorporation of serine in the place of alanine in peptidoglycan (Parveen and Reddy, 2017).

We have grown *P. fluorescens* SBW25 and our *serCGA* deletion mutants in serine rich (KB) and poor (M9) media, and compared cell phenotypes at various stages of growth (see new Figure 2—figure supplement 1). We have seen some intriguing cell phenotypes that we plan to investigate further – namely, very elongated long cells in the ∆*serCGA* mutants during growth in KB. However, these phenotypes differ from those reported by Zhang et al. for high-level serine toxicity in *E. coli* cells (Zhang and Newman, 2008).

3) Finally, recent experiments in our laboratory have shown broadly similar results to those reported here with a second, non-seryl tRNA type. Reduction of this second (glycyl) tRNA type leads to a fitness defect that is compensated by large, tRNA-encompassing duplications in a distinct region of the SBW25 chromosome. While the new work is preliminary and ongoing, it strongly indicates that our results are not limited to seryl-tRNAs.

To summarize: due to the combination of the three reasons above, we think it unlikely that serine toxicity is the primary cause of the ∆*serCGA* fitness defect in KB medium. However, serine toxicity may of course still contribute to the observed phenotypes. To reflect this development, we have included a figure of the cell morphologies (Figure 2—figure supplement 1), and a new paragraph in the relevant Results section.

2) Related to this point, in rich media it has been seen that charging levels for serine tRNAs can be very low at below 10% although serine concentration is high (Avcilar-Kucukgoze et al., 2016). This work is new to this reviewer, but this might be of interest to the authors and readers of the article.

Thank you for bringing this article to our attention! We have discussed the work in relation to our YAMAT-seq results (see point 1b above).

**References**

Burman JD, Harris RL, Hauton KA, Lawson DM, Sawers RG. 2004. The iron-sulfur cluster in the L-serine dehydratase TdcG from *Escherichia coli* is required for enzyme activity. *FEBS Lett***576**:442–444. doi:10.1016/j.febslet.2004.09.058

Cicchillo RM, Baker MA, Schnitzer EJ, Newman EB, Krebs C, Booker SJ. 2004. *Escherichia coli* L-serine deaminase requires a [4Fe-4S] cluster in catalysis. *J Biol Chem***279**:32418–32425. doi:10.1074/jbc.M404381200

Dittmar KA, Sørensen MA, Elf J, Ehrenberg M, Pan T. 2005. Selective charging of tRNA isoacceptors induced by amino-acid starvation. *EMBO Rep***6**:151–157. doi:10.1038/sj.embor.7400341

Goldman E, Jakubowski H. 1990. Uncharged tRNA, protein synthesis, and the bacterial stringent response. *Mol Microbiol***4**:2035–2040. doi:10.1111/j.1365-2958.1990.tb00563.x

Hauryliuk V, Atkinson GC, Murakami KS, Tenson T, Gerdes K. 2015. Recent functional insights into the role of (p)ppGpp in bacterial physiology. *Nat Rev Microbiol***13**:298–309. doi:10.1038/nrmicro3448

Metzger S, Schreiber G, Aizenman E, Cashel M, Glaser G. 1989. Characterization of the *relA1* mutation and a comparison of *relA1* with new *relA* null alleles in *Escherichia coli.J Biol Chem***264**:21146–21152.

Ott G, Schiesswohl M, Kiesewetter S, Förster C, Arnold L, Erdmann VA, Sprinzl M. 1990. Ternary complexes of *Escherichia coli* aminoacyl-tRNAs with the elongation factor Tu and GTP: Thermodynamic and structural studies. *BBA – Gene Struct Expr***1050**:222–225. doi:10.1016/0167-4781(90)90170-7

Parveen S, Reddy M. 2017. Identification of YfiH (PgeF) as a factor contributing to the maintenance of bacterial peptidoglycan composition. *Mol Microbiol***105**:705–720. doi:10.1111/mmi.13730

Plant EP, Nguyen P, Russ JR, Pittman YR, Nguyen T, Quesinberry JT, Kinzy TG, Dinman JD. 2007. Differentiating between near- and non-cognate codons in *Saccharomyces cerevisiae*. *PLoS One***2**:e517. doi:10.1371/journal.pone.0000517

Varenne S, Buc J, Lloubes R, Lazdunski C. 1984. Translation is a non-uniform process. Effect of tRNA availability on the rate of elongation of nascent polypeptide chains. *J Mol Biol***180**:549–576. doi:10.1016/0022-2836(84)90027-5

Winther KS, Roghanian M, Gerdes K. 2018. Activation of the stringent response by loading of RelA-tRNA complexes at the ribosomal A-site. *Mol Cell***70**:95–105. doi:10.1016/j.molcel.2018.02.033